# Time-course swRNA-seq uncovers a hierarchical gene regulatory network in controlling the response-repair-remodeling after wounding
Xinghai Yu [1,8], Jinghua Zhou[2,8], Wenkai Ye [2,8], Jingxiu Xu[2], Rui Li[3], Li Huang[1], Yi Chai[4], Miaomiao Wen[1], Suhong Xu [2,4] ✉ & Yu Zhou [1,5,6,7] ✉

Wounding initiates intricate responses crucial for tissue repair and regeneration. Yet, the gene regulatory networks governing wound healing remain poorly understood. Here, employing single-worm RNA sequencing (swRNA-seq) across 12 time-points, we delineated a three-stage wound repair process in *C. elegans*: response, repair, and remodeling. Integrating diverse datasets, we constructed a dynamic regulatory network comprising 241 transcription regulators and their inferred targets. We identified potentially seven autoregulatory TFs and five cross-autoregulatory loops involving *pqm-1* and *jun-1*. We revealed that TFs might interact with chromatin factors and form TF-TF combinatory modules via intrinsically disordered regions to enhance response robustness. We experimentally validated six regulators functioning in transcriptional and translocation-dependent manners. Notably, *nhr-76*, *daf-16*, *nhr-84*, and *oef-1* are potentially required for efficient repair, while *elt-2* may act as an inhibitor. These findings elucidate transcriptional responses and hierarchical regulatory networks during *C. elegans* wound repair, shedding light on mechanisms underlying tissue repair and regeneration.

Wounding is known to trigger a series of complex molecular responses necessary for repair and regeneration[1], such as activation of genes in stress response, tissue patterning, and cell growth, in multiple species, including planarians[2], sea anemones[3], hydra[4], axolotls[5], Drosophila[6], and mice[7]. However, the complete map of the wounding-induced genes and their regulators is still poorly understood[2,3]. Understanding the principles, organization, and functionality of the gene regulatory network (GRN) that controls wound repair and regeneration remains a critical problem. Moreover, the network modules and motifs in GRNs need to be decoded to understand gene regulation dynamics and organizing rules[8,9]. Recently, biologists started to identify the gene expression programs during wound repair by integrating multiple omics data in multiple organisms[10–12]. However, a systematic analysis of the gene regulatory network controlling the dynamic wound repair process is still lacking.

*C. elegans* free-living nematodes have a remarkable repair capacity that is mediated by direct actin polymerization[13]. Epidermal wounding induces an immediate response by releasing $Ca^{2+}$ signals from extracellular and intracellular to activate the wound repair process, including RHO-1 small GTPase activation, actin polymerization, and recruitment of key membrane repair proteins such as TSP-15, SYX-2, and EFF-1[14,15]. We also found intracellular organelle mitochondria could immediately respond to wounding by uptake $Ca^{2+}$ into the matrix, immediate fragmentation, and release of mitochondrial ROS production to promote wound repair[16]. Many oxidative genes and membrane repair-related genes can be upregulated in this process. In another process, epidermal wounding could trigger an innate immune response by inducing neuropeptide NLPs and CNCs expression, dependent on the p38MAPK and TGF-β signaling pathway[17]. Despite this scattered information regarding epidermal wounding in

[1]College of Life Sciences, TaiKang Center for Life and Medical Sciences, RNA Institute, Hubei Key Laboratory of Cell Homeostasis, Wuhan University, Wuhan 430072, China. [2]Center for Stem Cell and Regenerative Medicine and Department of Cardiology of the Second Affiliated Hospital, Zhejiang University School of Medicine, Hangzhou 310058, China. [3]Institute of Hydrobiology, Chinese Academy of Science, Wuhan 430072, China. [4]The Zhejiang University-University of Edinburgh Institute, 718 East Haizhou Rd., Haining, Zhejiang 314400, China. [5]Frontier Science Center for Immunology and Metabolism, Wuhan University, Wuhan 430072, China. [6]State Key Laboratory of Virology, Wuhan University, Wuhan 430072, China. [7]Institute for Advanced Studies, Wuhan University, Wuhan 430072, China. [8]These authors contributed equally: Xinghai Yu, Jinghua Zhou, Wenkai Ye. ✉e-mail: shxu@zju.edu.cn; yu.zhou@whu.edu.cn

*C. elegans*, the comprehensive understanding of the entire process, including the time-lapse gene activation and their distinct roles in wound repair remain elusive.

Multiple key questions about transcriptional wound responses in *C. elegans* remain unresolved. First, although we proposed an evolutionary conserved time-course model of wound response and repair[18], based on limited findings in *C. elegans* and relevant studies in other models[1], do the transcriptional responses have different stages after wounding? Second, what are the identities of the transcription factors (TFs) and their regulated genes that orchestrate wound repair? Third, which changes in TFs network activity are specific to the different stages, and when do these changes occur? Deciphering the transcriptional regulatory network following wounding remains a central challenge in understanding wound repair process across various types of injuries.

In this study, we addressed these key questions by combining multiple high-throughput experimental techniques and complementary computational approaches. We developed an improved single-worm RNA sequencing (swRNA-seq) method, and used it to examine both epidermal wounding and control groups across 12 distinct time points throughout the repair process. This comprehensive study yielded a large time-course RNA-seq dataset from 78 samples. Through wound-versus-control comparative analysis and impulse-based model analysis, we identified 3366 differentially expressed genes classified into seven categories with different characteristics in responding to wound repair. Interestingly, further clustering analysis revealed three interconnected stages we termed response, repair, and remodeling. Moreover, by integrating our time-course swRNA-seq data with the ENCODE ChIP-seq data of 283 TFs and 400 motifs of 371 TFs from the CIS-BP database, alongside the 60 ATAC-seq datasets obtained from *C. elegans* adults, we reconstructed the dynamic regulatory networks and identified 241 key TFs in regulating the sequential responses. Notably, we found that a set of putative core TFs formed cross-autoregulatory loops in response to wounding. Moreover, we uncovered that those TFs potentially interact with chromatin factors and form TF-TF regulatory modules via intrinsically disordered regions (IDR), probably to ensure robust programming of wound repair. Our findings established the landscape of the GRNs in wound repair and provided the molecular basis to understand the complex responses to wounding for tissue repair and regeneration.

## Results

### Time-course swRNA-seq assay for investigating epidermal wound repair in *C. elegans*

To dissect the orchestrated temporal transcriptome during the tissue repair, we punctured the young adult worms (12 h post L4 stage) with micro-injection needles at both anterior and posterior trucks (Wounded group, W). Subsequently, we performed a time-course single-worm RNA sequencing (swRNA-seq) in parallel with the unwounded worms (Unwounded group, UW). By employing a streamlined lysis buffer, we used an advanced swRNA-seq approach that improves on previous methods[19,20].

Given that the *C. elegans* epidermis hyp7 constitutes a single syncytium with 139 nuclei, detecting RNA transcription from individual nuclei at the wounds would prove unfeasible. Therefore, swRNA-seq enables a comprehensive survey of gene expression in epidermal wound response and repair. Across the 12 designed time points (0.25, 0.5, 1, 2, 4, 6, 10, 12, 14, 16, 18, 24 h(s) post wounding (h.p.w.)), we collected three or four worms per time-point as biological replicates for both the UW and W groups. Subsequently, we extracted the poly(A)$^+$ RNA from each sample separately for deep sequencing (Fig. 1a and Supplementary Data 1).

We processed the swRNA-seq data according to a standard pipeline (Fig. S1A) and found a high correlation between different replicates using the reads counts for all annotated genes, such as the samples at 1 h.p.w. (Fig. S1B). As expectedly, samples from the adjacent time points showed a higher correlation than those from non-adjacent time points (Fig. S1C). Comparing W to UW samples at individual time points, we performed differential gene expression analysis with DESeq2 and identified varying numbers of upregulated and downregulated genes (Fig. S1D–E). Upon comparing differentially expressed genes (DEGs) across all time points, we identified 8266 DEGs that were differentially expressed in at least one-time point (Fig. 1b and Supplementary Data 2). Among these, 5443 genes exhibited upregulation while 3833 genes showed downregulation. Interestingly, 1010 genes were found to be upregulated at the one-time point but down-regulated at another time point (Fig. 1b, line-linked genes).

By inspecting the RNA-seq signals in UCSC genome browser and comparing fold change between wounding/non-wounded for each replicate, we found the biological replicates are consistent on individual genes (Fig. 1c–e and Fig. S1F–K). We compared our swRNA-seq data (unwounded) with previous single-worm RNA-seq data (recently published in *BMC Genomics*) that are from adult worms in the uninfected control group (noted as SW_BMC here). We found that most of the expressed genes are significantly overlapped (Fig. S2A). The correlation analysis further revealed a strong positive correlation ($R = 0.86$) between the two datasets (Fig. S2B). These results indicate that our data are consistent with previous studies in unwounded conditions.

To evaluate our swRNA-seq on capturing transcription changes, we further performed bulk RNA-seq on wounded worms at 1 h.p.w., and systematically compared our swRNA-seq and bulk RNA-seq data. Notably, our wounded (W) and unwounded (UW) samples are clustered into two distinct groups and well correlated between replicates in both swRNA-seq (Fig. S1B) and bulk RNA-seq data (Fig. S2C). swRNA-seq captures slightly more expressed genes than bulk RNA-seq, and impressively, the expressed genes in 82% of swRNA-seq and 90% of bulk RNA-seq are captured in both methods, when requiring expressed genes to have an average FPKM > 1 in different replicates (Fig. S2D). For DEGs (foldchange >1.5 and FDR < 0.05), 54% of swRNA-seq and 52% of bulk RNA-seq are common (Fig. S2E). Furthermore, we compared the gene expression changes of all expressed genes, and observed a high degree of consistency between the two methods ($R = 0.8$). (Fig. S2F). Together, the results indicate that swRNA-seq can profile transcription changes well.

Moreover, we hypothesized that swRNA-seq has an advantage in synchrony over bulk RNA-seq, especially for the earlier time-point after wounding. We performed both swRNA-seq and bulk RNA-seq at 0.25 h.p.w. to further confirm this hypothesis. The replicates are well correlated in wound and unwound conditions, respectively (Fig. S2G–H), and the number of commonly expressed genes is also significant (Fig. S2I), as the results of 1 h.p.w. However, the number of DEGs from bulk RNA-seq is remarkably smaller than that from swRNA-seq (Fig. S2J). Specifically, swRNA-seq detected about four times more DEGs (3027) than bulk RNA-seq (760). Further correlation analysis of DEGs showed that many DEGs specific in swRNA-seq show non-significant differences in bulk RNA-seq (Fig. S2K), while common DEGs are well correlated (Fig. S2L). Together, these results provide direct evidence supporting our hypothesis that swRNA-seq excels at capturing more DEGs during the early stage of injury. While there are some power differences, the direction of effect is highly consistent between swRNA-seq and bulk RNA-seq, which provides confidence that DEGs identified are robust.

To further validate our swRNA-seq data, we performed RT-qPCR assays on approximately 21 representative genes at four distinct time points (0.25, 2, 4, and 24 h.p.w.). The results show consistent change patterns between swRNA-seq and RT-qPCR analysis during the wounding and repair process (Fig. S3). Together, RT-qPCR confirms the gene expression changes observed in swRNA-seq results.

The expression change patterns were consistent with previous studies, including several genes whose roles in wounding repair were well established. For instance, gene epithelial-fusion failure (*eff-1*) together with syntaxin-2 (*syx-2*) can promote both endoplasmic and exoplasmic membrane repair after wounding[15]. In this study, we found that the two genes were quickly induced after wounding, reaching the highest level at 0.5 h.p.w. and then starting to decrease at 2 h.p.w. (Fig. 1c, d). For neuropeptide genes, *nlp-28* and *nlp-29*, which encode the antimicrobial peptide (AMP), expressed in *C. elegans* epidermis upon wounding and infection[21], they reached the maximum expression at 4 h.p.w. (Fig. 1e). Together, these

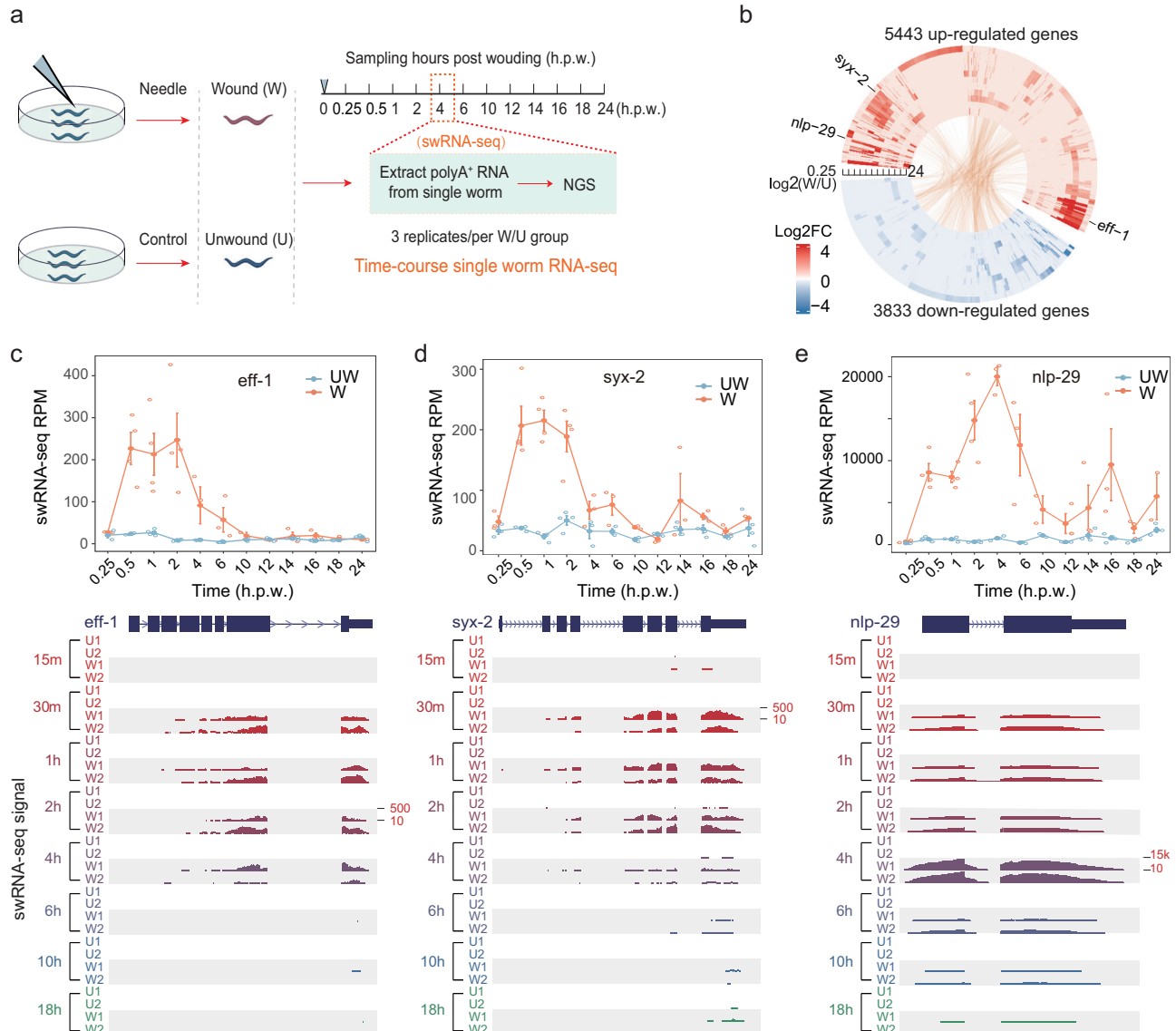

**Fig. 1 | Gene expression dynamics during epidermal wound repair by time-course swRNA-seq. a** Schematic diagram of time-course swRNA-seq assay for profiling gene expression dynamics during epidermal wound repair. Adult worms at Day 1 were randomly allocated into two groups, unwound (U) or wound (W) treated with a needle. From 0.25 h to 24 h after wounding, three worms were selected at each time point (12 in total), followed by single worm RNA-seq (swRNA-seq) of poly(A)$^+$ RNAs. **b** Circos diagram showing the differentially expressed genes (DEGs from the comparison between wound over unwound) at each time point. Each layer represents the DEGs at a specific time point from 0.25 h (outermost layer) to 24 h

(innermost layer) after wounding. The fold-changes of DEGs are represented as a color map from red (up-regulated) to blue (down-regulated). The same genes that are significantly up- or down-regulated at any two different time points are linked with lines inside the circus diagram. Quantified expression (top) and UCSC genome browser view of the swRNA-seq signals (bottom) for representative genes: *eff-1* (**c**), *syx-2* (**d**), and *nlp-29* (**e**). RPM number of reads per million, h.p.w hours post wounding, UW unwounded, W wounded. Tracks of wound conditions are highlighted with a gray background. See also Supplementary Figs. S1, S2 and S3.

findings not only validate previous reports but also provide further support for authentic identification of functional genes involved in the process of wound repair.

## A 3-stage sequential response and associated gene signatures in wound repair

To capture the dynamic trends of gene expression changes after wounding, we used ImpulseDE2, based on an impulse case-control model[22], to analyze gene expression trajectories in comparing W versus UW conditions (Fig. 2a). This method jointly analyzes data at all time points and is designed for identifying genes that change spanning multiple time points (Fig. 2a top) and filtering out other genes that only change at individual time points (Fig. 2a bottom). The impulse model considers the dependencies between time points compared to the single-time point significance approach, and

can distinguish transient change from permanent up- or down-regulation in time-course sequencing experiments. Under default parameters and an FDR cutoff at 0.01, we identified 8538 impulse-based differentially expressed genes (iDEGs). Interestingly, we found that those iDEGs could be clustered into three distinct groups based on the gene expression across the 12 time points after wounding, revealing three sequential stages: early (0.25 h), middle (0.5–4 h), and late (6–24 h) stages (Fig. 2b).

We further focused on highly expressed genes in the iDEGs (hiDEGs) satisfying the following criteria: (1) they fell within the top 75% of iDEGs when sorted by the sum of median gene expression values under W and UW conditions in decreasing order; (2) they exhibit a W/UW fold change of median expression values being larger than two in at least one time point. By applying these criteria, we identified 3366 hiDEGs and applied a k-means clustering technique to analyze their expression patterns. Utilizing the

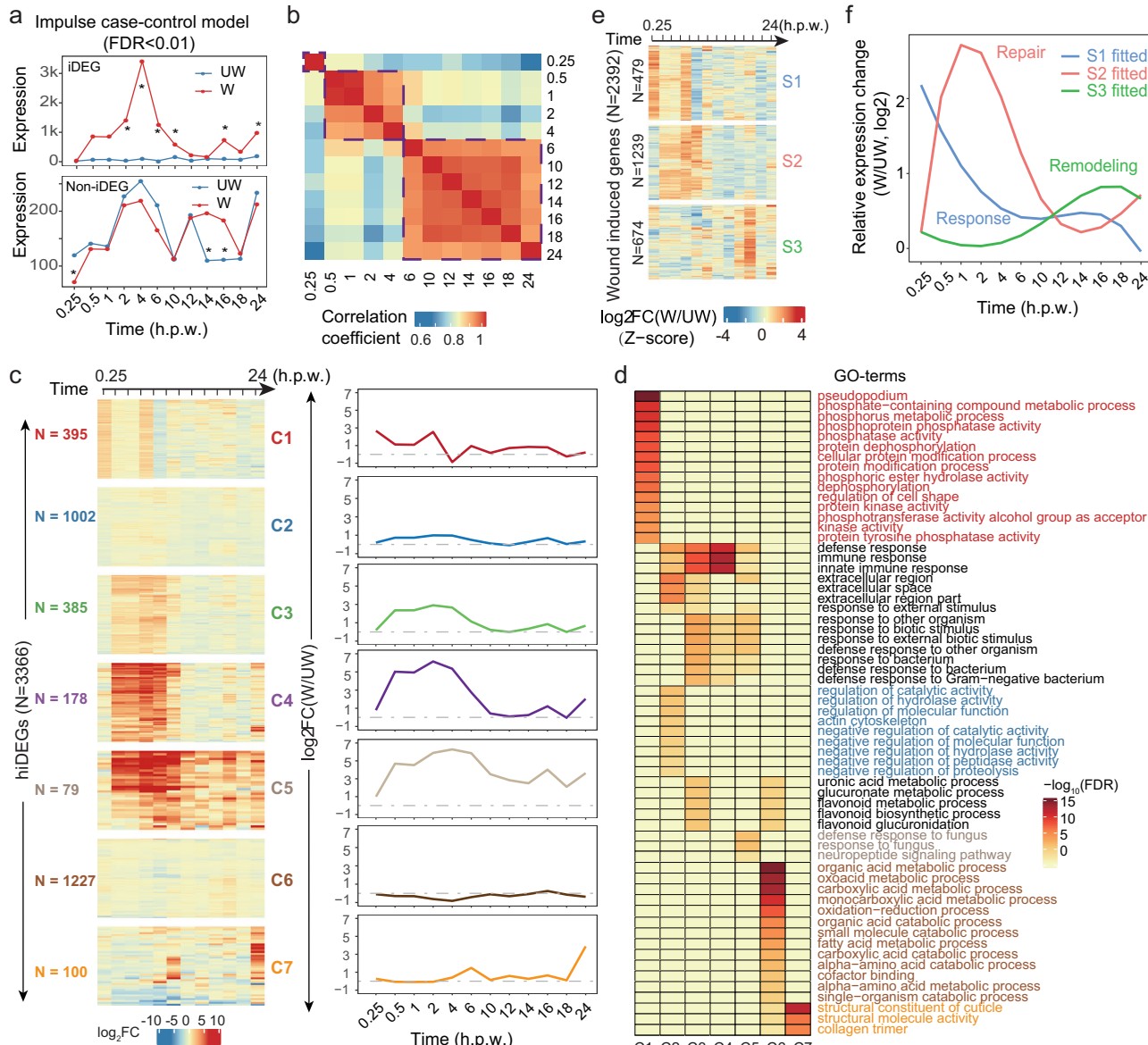

**Fig. 2 | Modules and stages of gene expression change after epidermal wounding.**
**a** Schematic diagram of impulse-based differentially expressed gene (iDEG) (top) and non-iDEG (bottom). The asterisk indicates significantly differential expression at an individual time point. **b** Heatmap and clustering of the correlation between time points using the mean expression values of 8538 iDEGs during wound repair. Dashed boxes highlight the three clusters of highly correlated time points. **c** Heatmap view (left) and composite view (right) of the expression dynamics for 7 clusters of genes during wound repair. The 3366 hiDEGs were classified into 7

clusters (C1-C7) according to their expression changes. The composite view represents the average fold-changes of all genes in a specific cluster. Log2FC: log2fold-change of wounded/unwounded. **d** Enriched GO-terms for the genes in at least one of the 7 clusters C1-C7. **e** Heatmap view of the expression dynamics of 2392 wound-induced genes for 3 stages (S1–S3). **f** Meta-profile of fitted relative expression changes for the three stages of wound-induced genes in **e**. The three stages are named Response, Repair, and Remodeling, according to the early to late stages of changes. See also Supplementary Figs. S4 and S5.

Elbow method, we determined that these genes could be effectively clustered into 7 distinct groups, denoted as C1-C7. These clusters were established based on their expression change profiles, providing insights into the underlying regulatory mechanisms governing gene expression. (Fig. 2c), the representative genes of which are shown in Fig. S4A. Our data analysis recovered *sta-2* as a hiDEG, the only transcription factor previously reported in epidermal wound repair of *C. elegans*[23], which was significantly induced at 0.5–4 h.p.w. (Fig. S4A, cluster 4). The result suggests that our data have the potential to reveal many more important regulators in the wound repair process.

To understand the functions of those hiDEGs, we performed gene functional enrichment analysis for all clusters separately and identified 62 GO-terms that are enriched in at least one cluster (Fig. 2d). Interestingly, the

genes in the C1 cluster, being immediately induced and reaching the highest expression at 0.25 h, significantly enrich terms related to phosphorylation and dephosphorylation (Fig. 2c, d), suggesting an important role of early signal transduction in wound repair. The genes in the C2-C4 clusters were slowly induced and reached the highest expression around 2 h, and they are mainly associated with innate immunity and defense response. Particularly, the genes in C2 are also enriched in extracellular space and actin cytoskeleton pathway, implying their roles in membrane repair. The C5 cluster is similar to C2-C4, slowly induced, but differs at later time points in maintaining upregulated status until 24 h, the genes in which are mainly associated with defense response and neuropeptide signaling pathway. The genes in C6, transiently down-regulated at 4 h, and the GO enrichment analysis revealed that C6 is predominantly associated with oxidation-

reduction processes and mitochondria (Fig. 2d). Finally, the genes in C7 were mainly activated at 24 h and are mainly enriched in cuticle and collagen (Fig. 2d). These findings provide a comprehensive understanding of the functions of hiDEGs across different stages of the wound repair process, highlighting key biological processes and pathways involved in epidermal wound repair.

Next, we defined a set of wound-induced genes from the hiDEGs by using a more strict criterion of requiring statistically significant two-fold up-regulation in at least one time point based on DESeq2 with an FDR cutoff at 0.05 in comparing W to UW samples. We identified 2392 wound-induced genes and performed k-means clustering based on the expression fold-changes (W/UW) of those genes. We used k = 3 for clustering as the three stages we revealed based on iDEGs (Fig. 2b), and found that the time points starting to change in the three groups coincide with the early, middle, and late stages (Fig. 2e). Subsequently we performed GO enrichment analysis for the genes in each group of S1–S3, respectively. Interestingly, the enriched GO-terms in the three groups or stages are similar to those enriched in the C1-C7 clusters based on hiDEGs: early (C1), middle (C2, C3, and C4), late (C6 and C7) (Fig. S4B). The clusters with the largest ratio of genes at each stages are: early (C1), middle (C3 and C4), and late (C7) (Fig. S4C), when excluding C2 and C6, which have the largest number of hiDEGs but with the least expression changes overall (Fig. 2C). Based on the common enriched GO-terms and the different modes of gene expression changes, we annotated the three stages as a Response (0.25 h), Repair (0.5–4 h), and Remodeling (6–24 h), sequentially (Fig. 2f). This comprehensive analysis provides deeper insights into the temporal dynamics and functional characteristics of the genes involved in epidermal wound repair.

Expectedly, we detected the activation of the two gene clusters *cnc-2* and *nlp-29* encoding antimicrobial peptides, which are known to be induced by different signal cascades after wounding[17,24] (Fig. S5A–D). Notably, we found that the genes in the same cluster are activated at different times with different scales and modes of expression changes, although their genomic loci are closely distributed, for the 6 genes in *cnc-2* cluster (Fig. S5A, C) and also the 6 genes in *nlp-29* cluster (Fig. S5B, D), suggesting a tight regulation and potentially different roles of the members in the same gene family cluster. Collectively, our analyses established a complete landscape of differentially expressed genes in wound repair and revealed a sequential gene expression program consisting of response, repair, and remodeling stages.

## A complete dynamic gene regulatory network for wound repair

To predict TFs and their inferred targets that might be involved in the wound response, we integrated our time-course swRNA-seq data with multiple datasets of transcription regulators. Specifically, we leveraged the ChIP-seq data of 283 *C. elegans* transcription factors from the ENCODE database, alongside 60 ATAC-seq datasets in *C. elegans* adults from the ChIP atlas database. Additionally, we incorporated the 400 motifs associated with 371 *C. elegans* transcription factors from the CIS-BP database. Through this integrative approach, we reconstructed gene regulatory networks aimed at elucidating the molecular mechanisms underlying wound repair in *C. elegans*. (Fig. 3a left). We used iDREM program, based on an Input-Output Hidden Markov Model, to identify the bifurcation points of genes sharing similar expression profiles and detect the transcription factors controlling the split[25]. Based on the foldchange matrix of 3366 hiDEGs across the time series and predicted TF-target interactions using ChIP-seq peaks and known TF motifs, we identified 61,267 candidate TF-target gene interactions placed into 10 paths (p1-p10), each of which represents a set of genes that display similar expression patterns under the inferred regulation of specific TF(s) at different bifurcation time points (Fig. 3a right and Supplementary Data 3). Those interactions include 241 unique TFs and 1781 unique inferred target genes. Of note, similar motifs can be bound by different TFs and these regulatory interactions between TFs and inferred targets are not from direct experiments such as ChIP-seq in the wound repair process, and merit further experimental validataion.

Notably, we revealed *jun-1* and *fos-1* in the gene regulatory network of *C. elegans* for the first time, although previous studies reported that they

were significantly induced after wounding in humans, mice, and Drosophila[26]. Moreover, their downstream targeted genes *W01F3.2* and *ifd-2*, which we found in the network, were also known to be targeted by JUN-1/FOS-1 in other species, including mice[26]. A recent study of wound repair performed RNA-seq of inner and outer wound regions in skin wound healing mouse model[11]. By reanalyzing its RNA-seq data, we identified 301 up-regulated TFs in response to skin injury in mice, and 92 homologous TFs of which in *C. elegans* were significantly up-regulated (Fig. 3b and Fig. S6A). As shown in Fig. 3c, the top 15 transcription regulators (TFs, cofactors, or signaling proteins) based on up-regulation foldchange at 2 h.p.w. in *C. elegans* are all up-regulated, including *jun-1*, *fos-1*, *cwn-1*, etc. The signaling protein WNT7B is found to be critical for kidney repair and regeneration in macrophages of mouse[27], and here, we observed the *C. elegans* Wnt gene *cwn-1* significantly up-regulated after wounding (Fig. 3c). The down-regulated genes in *C. elegans* also have significant overlap with those down-regulated ones in mouse skin wound healing (Fig. S6B–D). These results support the hypothesis of a common foundation of wound repair across phylogeny[1].

We next characterize the TFs and their possible targets in the inferred gene regulatory network by the response, repair, and remodeling stages. There are 206, 130, and 121 TFs in regulating 626, 1094, and 424 putative target genes in the three respective stages (Fig. 3d). The regulatory inter-actions can be classified into TF-(non-TF), TF-(other-TF), and TF-(self-TF) according to the type of TF targeted genes. Notably, there are 194, 75, and 39 transcription factors that can cross-regulate other transcription factors in the response, repair, and remodeling, respectively, indicating a hierarchical regulatory mode during wound repair (Fig. 3e). Interestingly, we also identified four and three TFs that potentially auto-regulate themselves at the response and repair stage, respectively (Fig. 3e). Using the degrees of nodes in the TF-TF subnetwork for the three stages, we identified many tran-scription factor hubs, such as ZIP-2 (degree 145) and SNPC-3.4 (degree 126) in the response stage (Fig. 4f), JUN-1 (degree 22) and FOS-1 (degree 17) in the repair stage (Fig. 4g), and SNPC-3.4 (degree 26) in the remodeling stage (Fig. 3h), respectively. Specifically, in the early response stage, we identified 33 transcription factors with altered mRNA expression, among which nine TFs were up-regulated while 24 TFs were down-regulated (Fig. S6E). We found that these TFs might form a three-layer hierarchical regulatory net-work by themselves (Fig. 3i), including a direct regulatory link from DIE-1 to LSY-27, consistent with the finding that either LSY-27 or DIE-1 mutation had a similar phenotype in asymmetric neuronal differentiation of *C. elegans*[28].

Meanwhile, we performed de-novo motif discovery in the promoter sequences of up-regulated genes at different time points using the Homer program[29]. The HIF-1, JUN-1, PQM-1, and CEBP-1 binding motifs were revealed to be significantly enriched (Supplementary Data 4). These factors are also recovered in our integrative analysis, which provides a much larger spectrum of TFs. Together, our integrative analysis based on time-course RNA-seq, TF ChIP-seq, ATAC-seq and TF motifs constructed a complete dynamic gene regulatory network of wound repair, being useful for *C. elegans* and also other species and meriting further experimental validation.

## Potential transcription factor autoregulatory loops during wound repair

We further characterized the autoregulatory loops in wound repair, a unique regulatory module with feedback being able to robustly and fast respond to stimuli, which was shown to maintain corneal epithelial homeostasis[30]. We identified 7 TFs that could regulate themselves by binding to their own promoter regions supported by TF ChIP-seq peak or motif occurrence, including NHR-76, ELT-2, SNPC-3.4, DIE-1, JUN-1, FKH-7 and NHR-178 (Fig. 4a, b and Fig. S7A–E). To infer their activating or repressing regulatory function on gene expression, we compared the TF mRNA expression level with its inferred unique targets at all time points during wound repair, and found that JUN-1 and NHR-178 could enhance the expression of their inferred unique target genes (Fig. 4c, d), while ELT-2 seemed to repress the expression of its inferred targets (Fig. S7F).

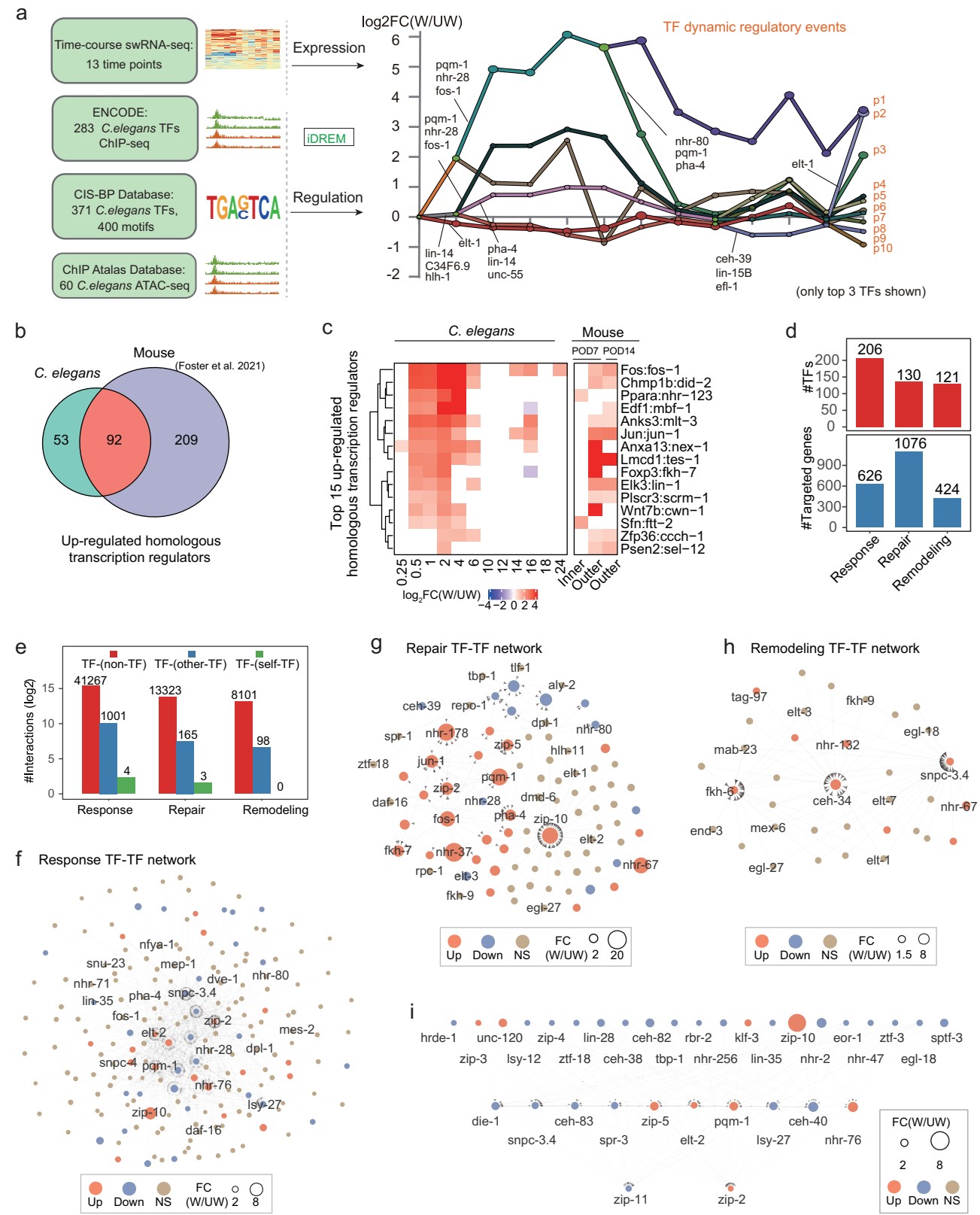

We noticed that the two TFs (*nhr-178* and *jun-1*) are the most strongly activated during wound repair (Fig. 4e) and are potentially regulated by the same set of five TFs, including three other TFs: FOS-1, PQM-1, and NHR-28 (Fig. 4a, b). We further analyzed their mRNA level during wound repair and found that four TFs (*fos-1*, *jun-1*, *pqm-1*, and *nhr-178*) were induced, whereas *nhr-28* mRNA was down-regulated at 0.5 h.p.w. (Fig. 4f). Protein-

protein interaction (PPI) analysis revealed that these transcription factors can interact with each other and may form a large protein complex as their binding sites are close in the genome (Fig. 4g and Fig. 4a, b). We further analyzed the expression changes of NHR-28's inferred unique target genes and found that NHR-28 may function as a repressor (Fig. 4h), consistent with its decrease during the activation of NHR-178 and JUN-1 at 0.5 h.p.w.

**Fig. 3 | The TF dynamic regulatory network for *C. elegans* wound repair.**
**a** Graphical illustration of the data and strategy for constructing the TF dynamic regulatory network. The integrated data include ChIP-seq data of 283 transcription factors obtained from ENCODE database, 400 motifs of 371 TFs obtained from CIS-BP database, 60 sets of ATAC-seq data obtained from the ChIP atlas database, and processed data from our time-course swRNA-seq data (left). The iDREM program integrated the expression and regulation data to construct a dynamic transcription factor (TF) regulatory network consisting of a series of TF-target gene regulatory events during wound repair (right). Each path represents a set of co-expressed genes. Split nodes represent a branch where a set of co-expressed genes begin to express differentially due to regulatory events. Only the top 3 TFs were shown on branches.
**b** Venn diagram showing the overlap of up-regulated homologous transcription regulators in mouse and *C. elegans* after epidermal wounding. The mouse data were

from ref. 11. **c** Heatmap showing gene expression changes for the top 15 overlapped up-regulated homologous transcription regulators as in **b**. The expression changes were calculated as the log₂ ratio of W (wound) to UW (unwound) conditions from *C. elegans* or mouse. **d** Statistics of the numbers of TFs (top) and inferred target genes (bottom) in the dynamic regulatory network for the three stages of wound repair. **e** Statistics of the numbers of regulatory events classified into three regulatory types TF-(non-TF), TF-(self-TF) or autoregulatory, TF-(other-TF) for the three stages. TF-TF subnetworks at the Response stage (**f**), Repair stage (**g**), and Remodeling stage (**h**). **i** TF-TF subnetwork for differentially expressed TFs at the Response stage. Each circle represents a TF, and the arrows represent the regulatory direction from TF to TF. The three colors (red, blue, and brown) represent up-regulated, down-regulated, and non-significant changes, respectively, and the circle size represents the log₂foldchange value. See also Supplementary Fig. S6.

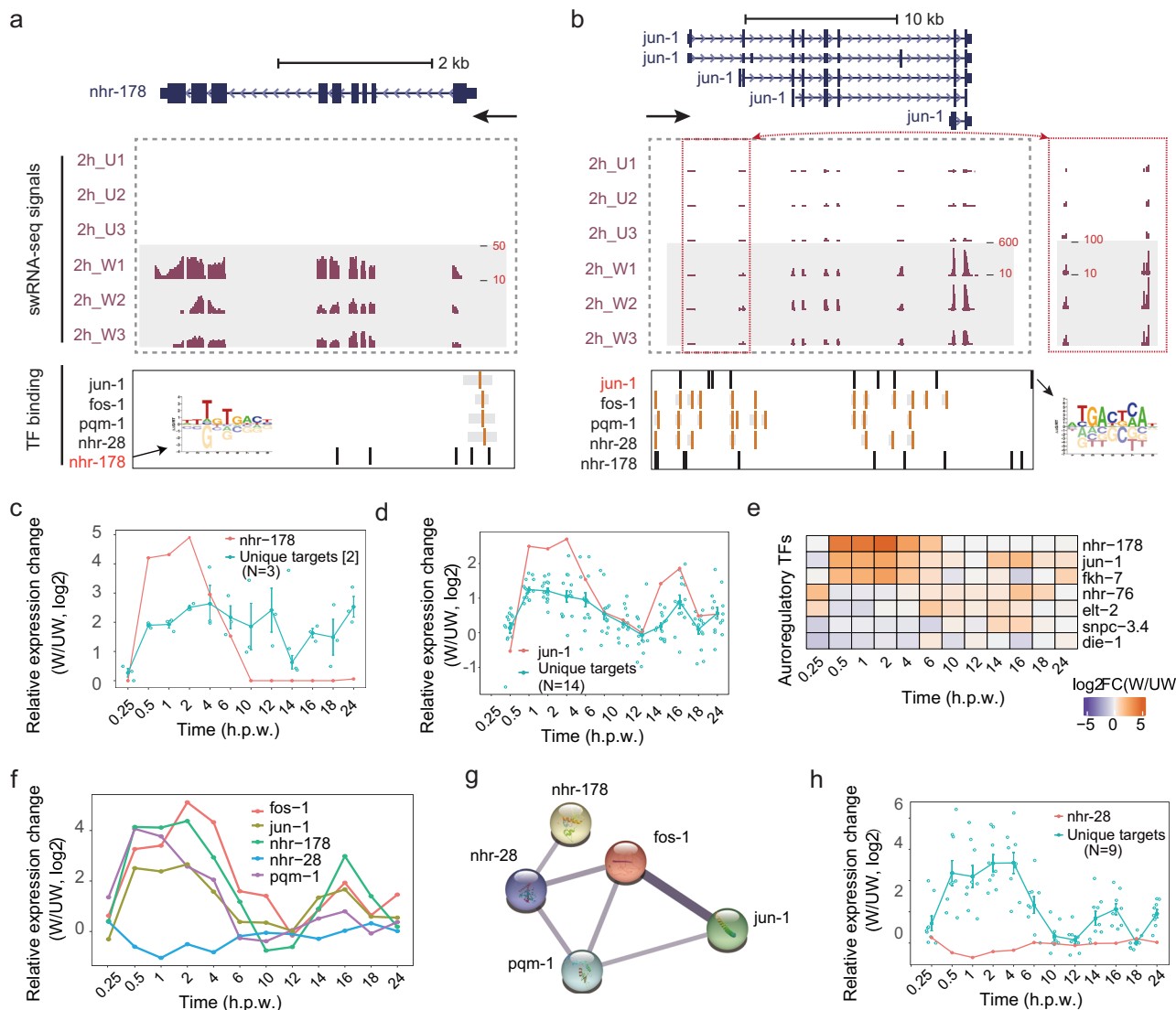

**Fig. 4 | Putative TF autoregulatory regulations during wound repair.** UCSC genome browser track views of the RNA-seq signals and TF binding data for *nhr-178* (**a**) and *jun-1* (**b**) during wound repair. The ChIP-seq peaks and motif occurrences of TFs are shown as orange and black lines, respectively. U1-U3: Unwound conditions; W1-W3: Wound conditions. Expression profiles of TF and its inferred unique targets during wound repair for *nhr-178* (**c**) and jun-1(**d**). The unique targets [2] were defined as those genes regulated by a maximum of two TFs including the

corresponding TF, respectively. **e** Heatmap showing the expression changes of putative autoregulatory TFs during wound repair. **f** Expression profiles of the shared TFs regulating both *jun-1* and *nhr-178* during wound repair. **g** Protein-protein interaction of shared TFs regulating both *jun-1* and *nhr-178*. **h** Expression profiles of TF and its inferred unique targets during wound repair for *nhr-28*. See also Supplementary Fig. S7.

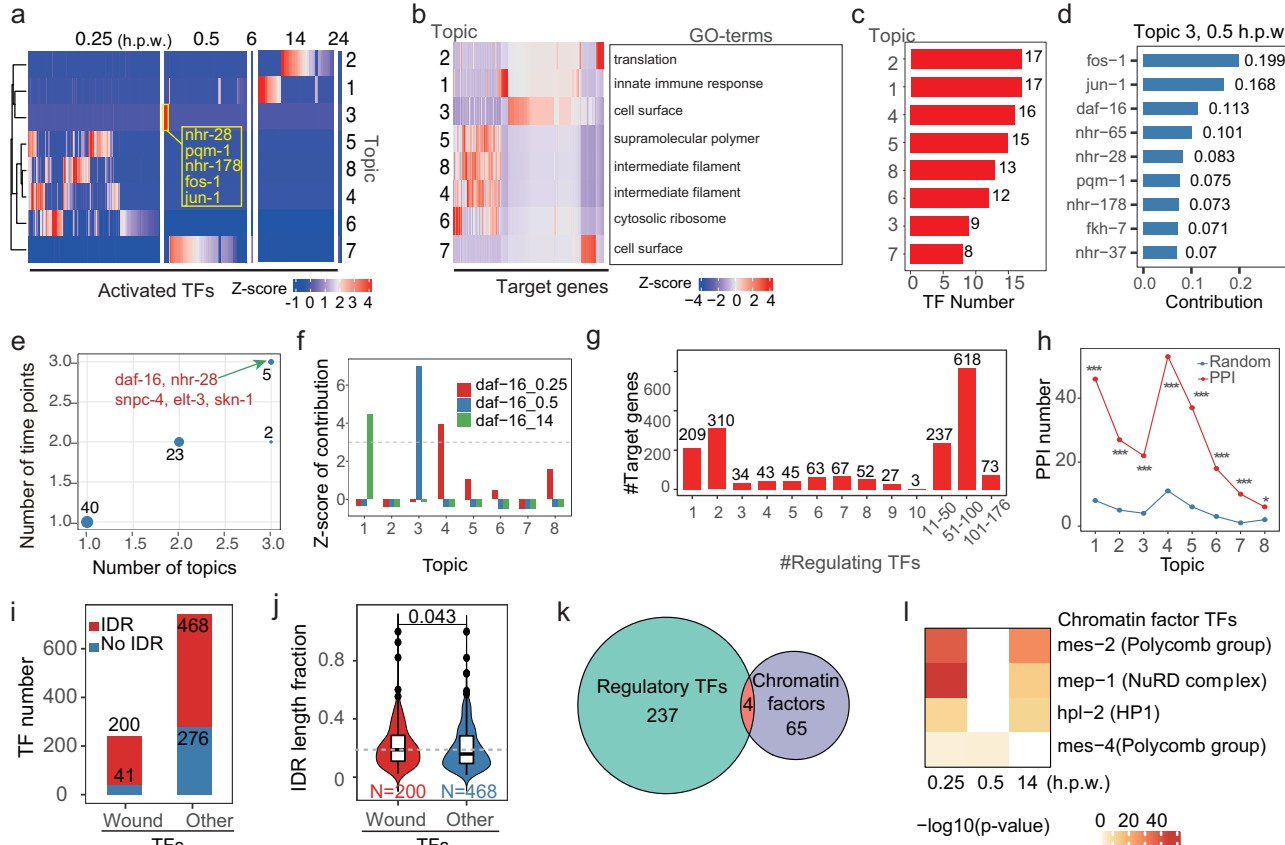

**Fig. 5 | TF combinatorial regulations during wound repair. a** Heatmap showing the relative contributions of the activated TFs in the 8 TF topics at four time points. The yellow box highlights the five TFs shown in Fig. 4g. **b** Heatmap showing the relative contributions of inferred target genes in the 8 TF topics. The top 1 enriched GO-terms for 8 TF topics are shown on the right. **c** Statistics of the number of TFs in the 8 TF topics. **d** Bar plot showing the contributions of the TFs in TF topic 3. **e** Statistics of the TFs by the number of occurrences in different TF topics and time-points. Representative TFs are labeled in red. **f** Relative contribution of TF DAF-16 in all topics at three time points. Gray dashed line represents the z-score cutoff at 2. **g** Statistics of the numbers of inferred target genes regulated by the specific numbers of combinatorial TFs. **h** Observed and expected protein-protein interaction (PPI) numbers between TFs in 8 TF topics. $*p < 0.05$, $**p < 0.01$, $***p < 0.001$. **i** Statistics of the numbers of TFs with IDR (red) or without IDR (blue) in wound-activated TFs and the remaining TFs. **j** Violin plot showing the length fractions of IDR regions to full-length in wound-activated TFs and the remaining TFs with IDR(s). The central line of the box plot represents the median value, and the lower and upper whiskers of the box represent the first and third quartiles, respectively. The upper whisker extends from the hinge to the largest value no further than 1.5 * IQR from the hinge (where IQR is the inter-quartile range). The lower whisker extends from the hinge to the smallest value at most 1.5 * IQR of the hinge. **k** Venn diagram showing over-lapped TFs between the 241 regulatory TFs in the regulatory network and 69 known chromatin factors in *C. elegans*. **l** Heatmap showing the significance of regulatory functions for the four overlapped TFs in **k** at three regulatory time points. See also Supplementary Figs. S8 and S9.

These results indicate that different TFs may form regulatory complexes and thus might more robustly respond to stimuli such as the epidermal wound.

## Differential regulatory modules by combinatorial TFs during wound repair

To systematically explore the combinatorial regulation of transcription factors during wound repair, we used a topic model-based method to identify TF regulatory modules or TF topics[31,32]. The combinatory TF modules or topics were inferred from the regulatory networks of TF-genes we identified using iDREM previously. Here, in topic model terminology, each inferred targeted gene was considered as a "document" while its regulating TFs were considered as "words", and a combination of TFs that co-regulate multiple genes were defined as a module or topic (Fig. S8A). We concatenated the gene-TF matrices at 0.25, 0.5, 6, 14, and 24 h.p.w as input and found 8 TF topics using the R package topicmodels (Fig. 5a, Fig. S8B–C, and Supplementary Data 5).

Expectedly, this systematic analysis recovered the combination of five TFs (*nhr-28*, *pqm-1*, *nhr-178*, *fos-1*, and *jun-1*) in Fig. 4, which were contained in topic 3 at 0.5 h.p.w. (Fig. 5a, yellow box). We performed GO-term analysis for the inferred target genes associated with the 8 topics, respectively, and the top GO-term of each was shown in Fig. 5b. The 8 topics

had different numbers of TFs, ranging from 8 (topic 7) to 17 (topic 2) (Fig. 5c).

Moreover, we counted the numbers of topics and time points that every TF participated in the 8 topics, and found that *daf-16*, *nhr-28*, *snpc-4* and *skn-1* contributed in three topics at three time points (Fig. 5e). DAF-16 was involved in the TF combinatorial regulations of topics 1, 3 and 4, and functioned at 14, 0.5 and 0.25 h.p.w., respectively (Fig. 5f). We analyzed the numbers of regulating TFs for all inferred target genes in our network, most of which were found to be potentially regulated by multiple TFs (Fig. 5g). These results suggest a potential combinatory regulation of multiple TFs in different stages of wound repair, meriting further investigation.

## PPI and intrinsically disordered regions potentially drive TF combinatorial regulation in wound repair

To understand the molecular basis of those combinatorial regulations, we analyzed the protein-protein interactions in each topic using the STRING database, and found that there were significantly higher frequencies of protein-protein interactions for the proteins in the same topic, compared to corresponding random controls (Fig. 5h). Furthermore, we analyzed the intrinsically disordered regions (IDRs) of those TFs functioning in *C. elegans* wound repair, as IDRs were found to be important for TF-TF interactions

and for TFs to exert biological functions and proteins with IDR can facilitate dynamic assembly of protein complex and function as signaling hubs[33,34]. We found that 200 of 241 TFs involved in wound repair have predicted IDRs, the proportion of which was significantly higher than other expressed TFs (Fig. 5i and Supplementary Data 6). The fractions of IDR length to protein total length were also significantly higher in TFs involved in wound repair than those in non-involved but expressed TFs (Fig. 5j), and this difference was also observed for wound TFs functioning in the response and remodeling stages of wound repair (Fig. S9A).

Next, we explored the relationship between IDR length and PPI frequency in TFs and found a weak positive correlation between them of all wound TFs (Fig. S9B), and this positive correlation was higher in TFs activated at 0.25 h.p.w. (Fig. S9C). Moreover, the IDR length and PPI frequency of TFs in all TF topics except topic 8 showed a stronger positive correlation (Fig. S9D). These results together suggest that the protein-protein interactions between TFs and the IDRs of those TFs may drive the extensive combinatorial regulations of TFs in wound repair.

Interestingly, we found that four of the 241 TFs were annotated as chromatin factors (Fig. 5k) and they (MES-2, MEP-1, HPL-2, and MES-4) were most significantly activated at 0.25 h post wounding (Fig. 5l). Using the protein-protein interactions in the STRING database, we found that the TFs in the response stage could interact with chromatin factors including histone methyltransferase HDA-1 interacting with LSY-27, DIE-1, RBR-2, and UNC-120 (Fig. S8D). These results suggest that transcription factors may form a complex with chromatin factors to modulate chromatin structure and thus regulate transcription in the early stage of wound repair.

To explore how the two TFs harmonize, we examined the RNA-seq data upon loss of two TFs simultaneously. We selected two TFs, *jun-1* and *pqm-1*, as representatives to perform RNA-seq experiments after wounding upon the loss of a single TF and both TFs. The correlations between the two replicates for all samples are good (Fig. 6a). In *pqm-1(ok485)*, 630 genes were down-regulated, while 451 genes were up-regulated (Fig. 6b). Notably, the expression of PQM-1 regulated genes in our network is significantly decreased in *pqm-1(ok485)* after wounding compared to non-target genes (Fig. 6c). Similarly, after RNAi knocking down of *jun-1*, 522 genes were up-regulated, and 159 genes were down-regulated (Fig. 6d). The expression of jun-1-regulated genes in our network significantly decreases compared to non-target genes (Fig. 6e). These results demonstrate that either JUN-1 or PQM-1 mainly promotes gene expression of its targets. Moreover, we examined the RNA-seq data upon loss of two TFs simultaneously (Fig. 6f). We found that the 218 common target genes of jun-1 or pqm-1 show a stronger decrease than 459 JUN-1 specific targets, 84 PQM-1 specific targets, and non-targets (Fig. 6g, h). Moreover, the common targets have stronger repression under the loss of both TFs than under the loss of either pqm-1 or jun-1 individually (Fig. 6i). This additive effect can be illustrated by the RNA-seq signal changes of two representative common targets, *nas-10* and *ttr-31*, under different conditions (Fig. 6j–k). Together, these results suggest the TFs function in a cooperative manner during wound repair.

### Validation of TFs newly discovered in the regulatory network

To further validate whether the TFs identified in the regulatory network play important roles in wound repair, we examined five TFs by monitoring the nucleoplasmic ratio change before and after wounding using a fluorescence reporter system. An increase of the nucleoplasmic ratio of a TF often indicates that it translocates to the nucleus to regulate transcription[35]. Our swRNA-seq analysis showed that NHR-76 mRNA was induced at 0.25 h.p.w. (Fig. 7a), although RT-qPCR validation using bulk RNA showed an insignificant increase (Fig. S3). Interestingly, we also observed that the nuclear level of NHR-76 protein significantly increased relative to its cytoplasmic level at 0.5 h.p.w. based on the GFP fluorescence imaging and quantification (Fig. 7b), suggesting NHR-76 be an important regulator. Similarly, another four TFs showed similar patterns including PQM-1 (Fig. S10A, B), FOS-1 (Fig. 7c, d), ZIP-2 (Fig. 7e, f), and PHA-4, an ortholog of the known pioneer transcription factor FoxA (Fig. 7g, h). Inspecting the

expression levels of their inferred unique targets at all time points, we found that the four TFs PQM-1, FOS-1, and ZIP-2 might play activating roles in transcription regulation (Fig. S10C–E).

A previous study showed that DAF-16 needs to form a complex with the chromatin remodeler SWI/SNF in binding to the promoter of its inferred targeted genes to promote stress resistance and longevity[36], while another study reported that DAF-16 could integrate the signals from different pathways to modulate aging and longevity[37]. Unexpectedly, we found that the mRNA expression level of *daf-16* did not change significantly (Fig. 7i), although we inferred that DAF-16 should be activated at the response and repair stage based on the regulatory network (Supplementary Data 5). This result of *daf-16* was further validated by RT-qPCR, consistent with the findings of swRNA-seq (Fig. S3). We inferred that DAF-16 might positively regulate its inferred target genes by inspecting the mRNA expression changes of its unique targets (Fig. S10F), which was further confirmed by RNA-seq in daf-16(mu86) worms. We found that the inferred targets of DAF-16 show a significant decrease in unwounded and wounded conditions (Fig. S10G, H). These results demonstrate that there is a regulatory relationship between the predicted TF and target genes. Indeed, we observed that DAF-16 protein was translocated from the cytoplasm to the nucleus in less than 2 min after wounding (Fig. 7k) and kept in the nucleus up to at least 1-h post wounding (Fig. 7l), consistent with the previous report that DAF-16 can quickly translocate to the nucleus through its phosphorylation[38]. Similarly, *nhr-76* had little expression change at 0.5 h (Fig. S3) but showed protein translocation from the cytoplasm to the nucleus. These results indicate that TFs can be regulated at both transcriptionally and post-transcriptionally levels to respond to stimuli such as epidermal wounding quickly.

To identify whether the TFs identified in the regulatory network are required for epidermal wound repair, we performed RNAi knockdown of *nhr-76* and *daf-16* in the adult epidermis specifically as previously described[14]. We observed more delayed wound healing at 3 h.p.w. and especially at 6 h.p.w. under *nhr-76* and *daf-16* knockdown compared tocontrol (L4440), as shown by the larger diameter of wound sizes (Fig. 7m, n). To further evaluate the effects of *nhr-76* and *daf-16* knockdown on membrane damage repair, we stained wounded animals with membrane-impermeable dye trypan blue (TryB) and found that loss of *daf-16* or *nhr-76* causes significantly larger TryB positive staining percentages than the negative control (Fig. 7o), suggesting that *daf-16* or *nhr-76* promotes membrane repair. These results demonstrate that the newly discovered regulators DAF-16 and NHR-76 play important roles in epidermal wound repair. We extended our functional validation by performing trypan blue staining assay at 6 h.p.w. for 18 TFs by RNAi knockdown. We calculated the trypan blue staining positive ratios and found four TFs have significant differences in this phenotype compared to controls (Fig. S11A). Specifically, RNAi knockdown of *elt-2* was found to enhance epidermal wound repair, whereas knockdown of *nhr-84* and *oef-1* was found to delaywound repair (Fig. S11A). Their functions are further confirmed by examining wound healing at 3 h.p.w. and 6 h.p.w., as illustrated in Fig. S11B and by statistics (Fig. S11C). These results highlight the potential of our wound repair regulatory network in identifying key candidates and providing mechanistic insights for further studies.

### Discussion

Understanding the gene regulatory network regulating the wound repair process is crucial for identifying potential therapeutic targets. Our results established the dynamic gene expression profiles across the whole wound repair process using swRNA-seq. We identified 8266 differentially expressed genes and 241 TFs as putative regulators during wound repair. By integrating genome-wide TF binding data and swRNA-seq, we constructed the first gene regulatory network of wound repair in *C. elegans*, and revealed diverse regulatory modes including autoregulatory, combinatorial, and hierarchical interactions, many of which are probably mediated through their IDR regions to form TF-TF complexes. Of note, we used the default thresholds in aiming to identify all putative factors and to establish a

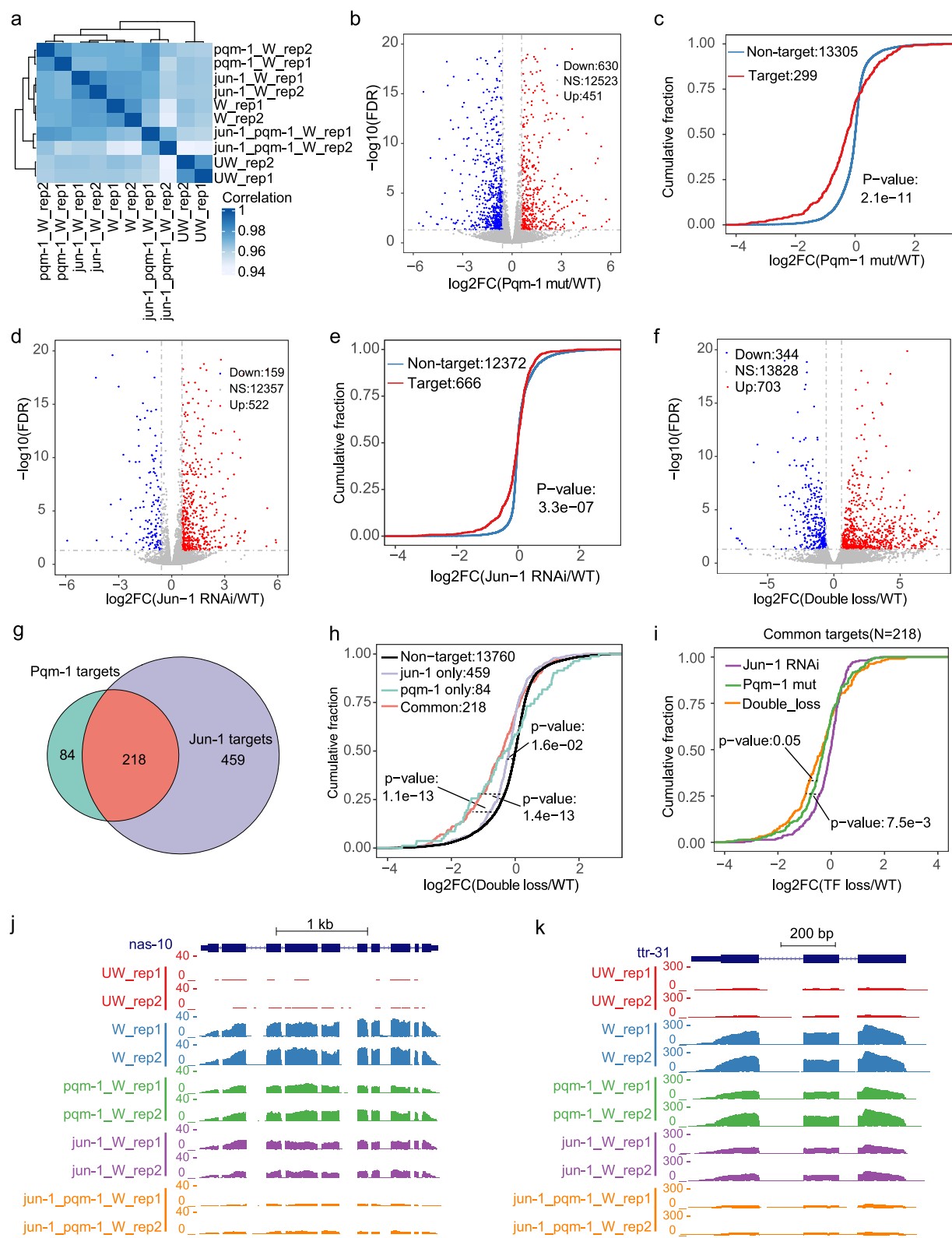

complete map for further studies. More stringent cutoffs can be used to identify the factors with more significant changes, and experimental validations are necessary to identify direct regulators and understand their functional roles in wound repair.

Wenemoser et al. have shown that three temporal waves of genes were activated during wound repair in planarians using expression microarray analysis[2], and we formally defined a sequential process consisting of response, repair, and remodeling stages in epidermal wound repair, which may generally be applicable to other stimuli or in other species. The classic injury response process in humans consists of three stages, including inflammation, new tissue formation, and remodeling stage[1], which could be correspondingly matched to the three stages we found in *C. elegans*. Some of

**Fig. 6 | Validation of TF effects on potential targets by TF inactivation. a** Heatmap of correlations in different TF inactivation conditions. **b** Volcano plots showing the distribution of differentially expressed genes in *pqm-1(ok485)* versus controls at 2 h.p.w. **c** Cumulative distribution of the expression changes of PQM-1's target (red) and non-target (blue) genes in comparing *pqm-1(ok485)* versus WT RNA-seq data at 2 h.p.w. The *p*-value was calculated by one-sided Wilcoxon rank-sum test. **d** Volcano plots showing the distribution of differentially expressed genes in jun-1 knockdown versus control at 2 h.p.w. **e** Cumulative distribution of the expression changes of JUN-1's target (red) and non-target (blue) genes in comparing jun-1 knockdown versus control RNA-seq data at 2 h.p.w. The *p*-value was calculated by one-sided Wilcoxon rank-sum test. **f** Volcano plots showing the distribution of

differentially expressed genes in *pqm-1(ok485)* under *jun-1* RNAi versus control at 2 h.p.w. **g** Venn diagram showing PQM-1 and JUN-1 targets in our regulatory network. **h** Cumulative distribution of the expression changes of the common target (red) and non-target (black) genes in comparing *pqm-1(ok485)* under *jun-1* RNAi versus WT RNA-seq data at 2 h.p.w. The *p*-value was calculated by one-sided Wilcoxon rank-sum test. **i** Cumulative distribution of the expression changes of common target genes under the conditions of single TF loss or double TFs loss at 2 h.p.w. The *p*-value was calculated by one-sided *t*-test. The UCSC genome track view showing the differences in the RNA-seq signals of *nas-10* (**j**) and *ttr-31* (**k**) under different conditions.

the associated biological events at different stages are common between humans and *C. elegans*; however, some are also species-specific events, such as platelets, cell migration, and angiogenesis in humans, which are not observed in *C. elegans*.

In addition, we found that FOS-1 and JUN-1, previously reported as immediate early response (IER) genes, were actually activated at the mRNA level in the repair stage, lagging behind the response stage. We revealed many more IER factors after wounding, through transcriptional regulation (including UNC-120, NHR-76, PQM-1, and ELT-2) or posttranscriptional regulation (including DAF-16) by phosphorylation-dependent transloca-tion into the nucleus.

A previous study has shown that JUN/AP-1 binding to a conserved 8 bp nucleotide sequence (TRE) positively autoregulates its own gene in HeLa and HepG2 cells[39], and we found that this mechanism is conserved in *C. elegans* at least in the epidermal wound repair process. Another study found that NHR-178 could interact with its own promoter by yeast one-hybrid (Y1H) assays[8], consistent with our finding that NHR-178 forms a putative autoregulatory loop during wound repair.

The multifaceted roles of ELT-2 in *C. elegans* development and stress responses are increasingly recognized. A recent study indicates that ELT-2 can function as a negative regulator in the intestine during embryo devel-opment in *C. elegans*, where it represses its expression across multiple developmental stages, as demonstrated by reporter experiments[40]. Inter-estingly, a previous study shows that ELT-2 activates its mRNA expression when exposed to heat shock, as evidenced by promoter-reporter experiments[41]. Another report highlights ELT-2's essential role in early immune responses against pathogens and recovery from *S. enterica* infection[42]. Here, our findings add to this complexity by suggesting that ELT-2 may play similar roles during wound repair, potentially forming an autoregulatory loop and repressing the expression of its target genes. The observed inhibition of epidermal wound repair following *elt-2* RNAi knockdown supports this notion. These results indicate that ELT-2 has broad functions during development and under stress conditions, probably mediated through a non-cell autonomous effect or interactions with other transcriptional factors, including chromatin factors such as MRG-1 (Fig. S8D).

How TF senses the damage and sequentially activates the immediate wound response genes is fundamental in the wound healing field. In this study, we found that TF may interact with chromatin factors through protein-protein interactions to open chromatin structure in the early stage of wound repair. Moreover, during the process of wound repair, TFs could form different regulatory modules to activate wound response genes by their IDR regions through a mechanism of liquid-liquid phase separation (LLPS). Our study provides dozens of regulatory TFs, and the molecular mechan-isms of wound healing including the roles of LLPS merit further studies. Importantly, it is necessary to perform ChIP-seq of those TFs in parallel with swRNA-seq upon wound repair to investigate their direct targets and reg-ulatory functions.

## Method and materials
### *C. elegans* strains that are used in this study
All the strains in this study are listed in Supplementary Data 7.

### *C. elegans* maintenance
All *C. elegans* strains were cultured at 20 °C on standard OP50 seeded Nematode growth media (NGM) plates, unless otherwise indicated.

### Needle wounding
N2 larvae at the L4 stage were transferred to new NGM plates seeded with OP50 bacteria and incubated at 20 °C for 12 h before the initiation of wounds. Following incubation, synchronized young adult N2 worms were punctured at both the 1/3 anterior and posterior sections of the body using a glass microinjection needle, taking care to avoid the gonadal region. The needle should enter the animal's body at an approximately perpendicular angle to the skin with an appropriate depth while ensuring no fatal harm is inflicted on the worm[43].

### Laser wounding
Worms were mounted on 10% agaroe pads in M9, in 12 mM levamisole. The epidermis of worms was wounded by a Micropoint UV laser, for examining DAF-16 translocation in Fig. 7k.

### Single worm RNA-sequencing
For each time point after wounding, normally three wounded and three unwounded worms were separately single transferred to 2 μl lysis buffer for RNA release. In all, 12 sequential time points were repeated for wounding, including 15 min, 30 min, 1 h, 2 h, 4 h, 6 h, 10 h, 12 h, 14 h, 16 h, 18 h, and 24 h after wounding.

We collected the single wounded or unwounded worms into lysis buffer, respectively. Single worm was picked promptly by worm pick to put into the lysis buffer of 1.5 ml tube. Then, all the liquid was removed from the tube under the anatomical lens by using a micropipette. An equal volume of lysis buffer was re-added into the tube and then kept on ice immediately. An improved blue pestle inside a 200 μL PCR tube with no lid was used as a rod to grind up a single worm under the anatomical lens rapidly.

The single worm was ground up in the lysis buffer. For each single worm lysate, 1 μl of each oligo-dT primer (10 μM) and dNTP (10 mM) was added to the PCR tube and heated at 72 °C for 3 min then cooled for 2 min. 6 μl reverse transcription mixture (100 U SuperScript II reverse tran-scriptase (Takara), superscript II first-strand buffer, 1 U RNAase inhibitor (Vazyme), 10 M betaine (Sigma), 6 mM MgCl$_2$ (Ambion), and 100 μM TSO primer) were then added directly and incubated using the following thermal cycle: 90 min at 42 °C, 15 min at 72 °C, and hold at 4 °C. The cDNA samples were amplified with 10 μl KAPA HiFi HotStart ReadyMix (Kapa Biosys-tems) and 12.5 μl 10 μM IS PCR primers. The purified cDNAs were frag-mented by TruePrep DNA Library Prep Kit V2. Different libraries were barcoded and pooled together followed by sequencing on Illumina (Vazyme Inc) Hiseq X 10 system. The total reads numbers of all samples are available in Supplementary Data 1.

After single worm lysis, The RT reaction was performed at 42 °C for 90 min and added dNTPs, tailed oligo-dT oligonucleotides, template-switching oligos (TSOs), betaine, magnesium chloride, and RNase inhibitor. The free dNTPs were added in the first step to improve the stabilization of RNA-primer hybridizations in RT-PCRs. Betaine (N,N,N-trimethylgly-cine) was added to the RT reaction to increase the thermal stability of reverse

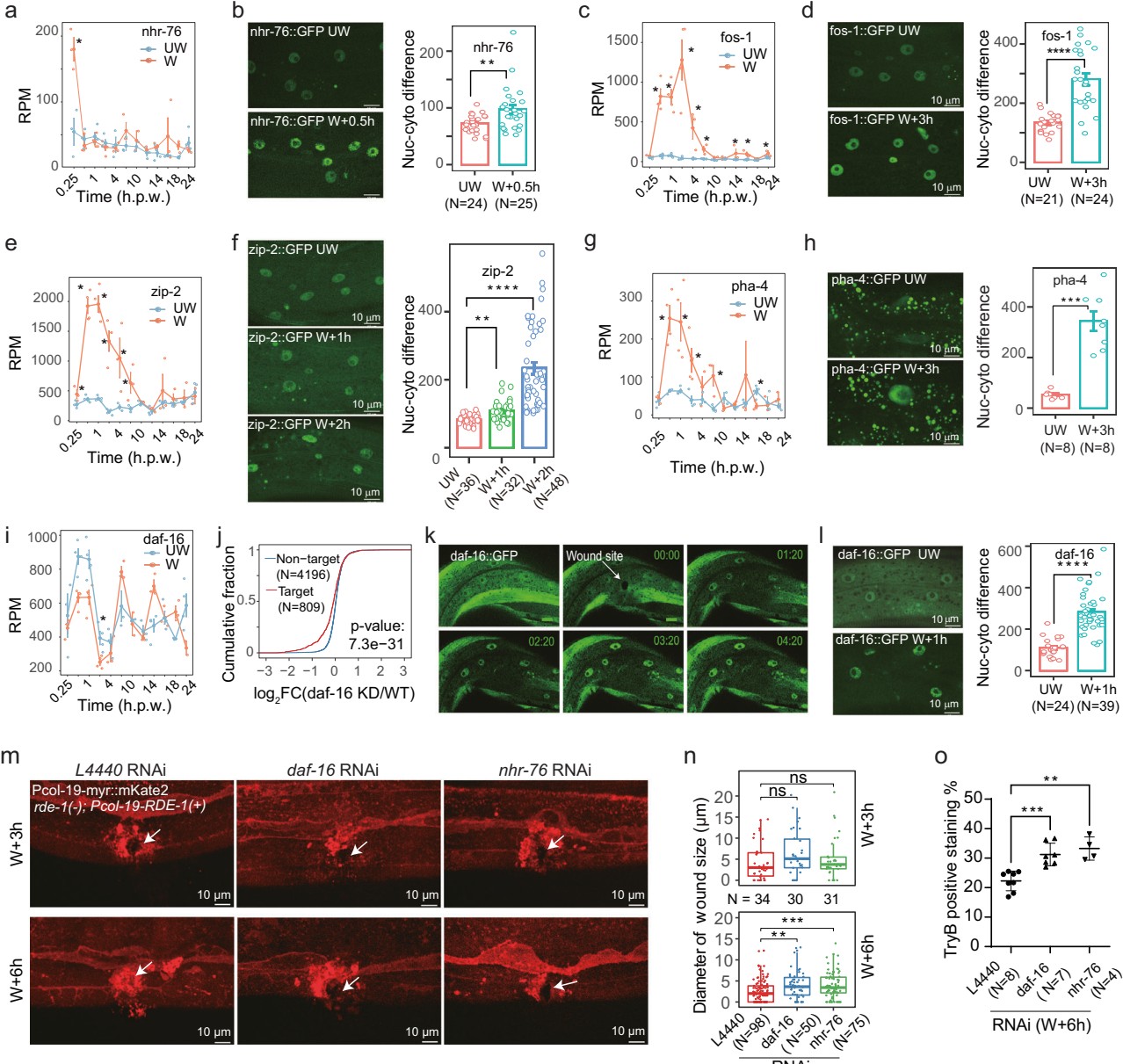

**Fig. 7 | Validation of uncovered regulatory TFs in wound repair. a** Dynamic expression profile of *nhr-76* under unwounded (UW) and wounded (W) conditions. **b** Fluorescence confocal images showing NHR-76 translocated into the nucleus at 0.5 h post wounding (left) and the quantified Nuc-cyto relative fluorescence intensity (nucleus-cytoplasm) of NHR-76 in UW and 0.5 h.p.w. conditions (right). Dynamic expression profile (**c**), translocation imaging (**d**, left), and the nuc-cyto relative fluorescence intensity (**d**, right) for FOS-1, as **a**, **b**, respectively. Dynamic expression profile (**e**), translocation imaging (**f**, left), and the nuc-cyto relative fluorescence intensity (**f**, right) for ZIP-2, as **a**, **b**, respectively. Dynamic expression profile (**g**), translocation imaging (**h**, left), and the nuc-cyto relative fluorescence intensity (**h**, right) for PHA-4, as **a**, **b**, respectively. **i** Dynamic expression profile of *daf-16* under unwounded (UW) and wounded (W) conditions. **j** Cumulative distribution of the expression changes of DAF-16's target (red) and non-target (blue) genes in comparing *daf-16* knockdown versus WT microarray data. The *p*-value was calculated by one-sided Wilcoxon rank-sum test. Fluorescence confocal images showing DAF-16 quickly translocated into the nucleus in less than 2 min (**k**) and kept up to at least 1 h post wounding (**l**). **l** Nuc-cyto relative fluorescence intensity

(nucleus-cytoplasm) of DAF-16 in UW and 1 h post wounding conditions as left. **m** Representative confocal images of the wounded membrane in *L4440*, *daf-16*, and *nhr-76* RNAi animals at 3 h.p.w. (top) and 6 h.p.w. (bottom). The epidermal-specific RNAi strain P*col19*-myr::mKate2;*rde-1*(-);P*col-19*-RDE-1 was used for RNAi treatment and needle wounding. White arrows indicate wound sites. Scale bars, 10 μm. **n** Quantified diameters of wound sizes in *L4440*, *daf-16*, and *nhr-76* RNAi animals at 3 h.p.w. (top) and 6 h.p.w. (bottom). The *p*-value was calculated by one-sided Wilcoxon rank-sum test. The central line of the box plot represents the median value, and the lower and upper whiskers of the box represent the first and third quartiles, respectively. The upper whisker extends from the hinge to the largest value no further than 1.5 * IQR from the hinge (where IQR is the inter-quartile range). The lower whisker extends from the hinge to the smallest value at most 1.5 * IQR of the hinge. **o** Quantification of TryB positive staining of wounded worms at 6 h.p.w. Error bars represent the mean value ± SD. The *p*-value was calculated by one-sided Wilcoxon rank-sum test. *$p < 0.05$, **$p < 0.01$, ***$p < 0.001$. * indicated a significantly differential expression of the corresponding TF at the indicated time point after wounding. See also Supplementary Figs. S10 and S11.

transcriptase. The oligo-dT oligonucleotides containing 55-nt sequence included 30-nt poly-dT and 25-nt universal 5′ anchor sequence. Other chemicals were added to improve the enzyme activity and promote reverse transcription efficiency. The primers TSO carrying three riboguanosines at its 3′ end are added to the 5′ end of the mRNA during the RT reaction. Consequently, there is a 25-nt universal 5′ anchor sequence at both ends of cDNA. Thus, the universal sequence was created for secondary amplification.

After the RT reaction, KAPA HiFi DNA polymerase and ISPCR oligos were used in the first PCR amplification. By using ISPCR oligo primers, The amount of cDNA was exponentially amplified for subsequent fragmentation. The cDNA products should be evenly distributed as a smear between 0.25 kb and 2.0 kb. After purification, 1 ng cDNA was randomly fragmented by using enzymes. The second PCR amplification was performed using adapter-ligated fragments. In this reaction, the same universal adapter was used together with the index reverse adapter. These index primers adapter barcode each library, so that different libraries can be pooled together for subsequent sequencing.

Each library was purified by using VAHTS DNA Clean Beads before sequencing. In the purification steps, the library size was selected by regulating the ratio of DNA Clean Beads for sequencing on different types of Illumina platforms. In this protocol, the ratio of DNA Clean Beads was selected for the Illumina HiSeq X10 instrument. Different libraries were pooled together and sequenced on the Illumina system, and we normally sequenced 8 million reads per worm.

### RT-qPCR
RNA samples were prepared from 100 unwound or wounded worms at four time-points after wounding, 0.25 h, 1 h, 2 h, and 24 h, respectively, by using SteadyPure Universal RNA Extraction Kit (Accurate Biotechnology (Hunan) Co., Ltd). 500 ng RNA was converted to cDNA using Evo M-MLV RT Mix Kit with gDNA clean for Qpcr Ver.2 (Accurate Biotechnology (Hunan) Co., Ltd). The cDNA (5 ng) was then amplified in duplicate reactions with 2x ChamQ Universal SYBR qPCR Master Mix and 0.5 mM of each primer pair for the gene of interest and *rbd-1* using the Bio-Rad system. The qPCR experiment was repeated for 3–4 times.

The qPCR primers for the gene of interest were designed by Primer-BLAST tools using a standard method (https://www.ncbi.nlm.nih.gov/tools/primer-blast/index.cgi?).

### Bulk RNA-seq
For worms wounded for 15 min, 15 worms were needle wounded for 2 min as a unit. After repeatedly wounding 15 worms 8–10 times, the samples were sequentially collected into 200 μl Trizol lysis buffer. For worms wounded for 1 or 2 h, 40 worms were needle wounded for 5 min as a unit. After repeatedly wounding 40 worms 3 times, the samples were sequentially collected into 200 ul Trizol. 10–15 3 mm ceramic grinding beads were added. The 1.5 ml EP tube wa placed on the grinder, shaked at 30 Hz for 15 s, placed on ice for 30 s, and repeated 2 times. The sample-Trizol solution was transferred into a new RNAse-free 1.5 ml EP tube and quickly frozen in liquid nitrogen. The samples were stored at −80 °C until being sent to Novogene company for library construction and mRNA sequencing.

### Confocal imaging
Transgenic worms overexpressing GFP-fused transcription factor were wounded for 15 min, 30 min, 1 h, 2 h, and 4 h, respectively. SHX422 strain was wounded for 3 and 6 h. The wounding area was imaged by a Z-stacked image program using a spinning disk confocal microscope (Andor 100X, NA 1.46 objective) with excitation on 488 nm and emission on 525 nm filter. The Z step was set as 0.5 μm.

### Epidermal-specific RNAi
To conduct epidermal-specific RNAi knockdown, strains of CZ14540 and SHX422 were bleached for synchronization. The eggs were then seeded onto

RNAi bacteria, which is the recombinant HT115 E. coli overexpressing dsRNA of TFs. The RNAi specificity was checked by sequencing, and the RNAi efficiency was confirmed by blister phenotype after feeding duox-2 RNAi bacteria. When RNAi knocking down *jun-1* for RNA-seq, wide-type N2 worms were used instead following the same aforementioned method. *C. elegans* was then needle wounded at the young adult stage.

### Trypan blue staining and imaging
Trypan blue (TryB), a membrane-impermeable dye, was utilized to label the damaged membranes. Groups of at least 50 wounded young adult worms were submerged in a trypan blue staining solution (0.5%, w/v) for 1 h at 20 °C. Subsequently, the staining solution was washed off with M9 buffer until the solution cleared. Images were captured using a CCD camera (Nikon Digital Sight DS-Vi1). For statistical analysis, wounds exhibiting dark-blue staining were classified as TryB positive, while those with faded staining were considered TryB negative. The percentage of TryB-positive wounds was calculated to assess the efficiency of wound repair[44].

### Image quantification and statistical analysis
The absolute intensity of nuclei and body was analyzed by ImageJ (https://imagej.nih.gov/ij). The relative intensity of nuclei was measured by the equation: Relative intensity(nuclei) = Absolute intensity(nuclei)—Absolute intensity(body). All statistical analysis used GraphPad Prism9 (La Jolla, CA). The Standard Error of the Mean (SEM) was used as the y-axis error bars for bar charts plotted from the mean value of the data. One-way ANOVA and Dunnett's multiple comparison tests were used to detect the relative intensity changes after wounding.

### swRNA-seq data analysis
Clean reads were pre-processed and firstly mapped to rRNAs and tRNAs, and those unmapped reads were then mapped to the *C. elegans* genome (version: ce11) with gene annotation WS258 using STAR aligner[45] version 2.5.3a with the following settings: "—outFilterMatchNmin 40". The featureCounts program from Rsubreads package[46] was used to counts of reads mapped to each gene and the counts data normalized by sequencing depths were used in the downstream analyses.

Differentially expressed genes (DEGs) at each time point between Wounded group and Unwounded group were defined by calling DESeq2[47] on the counts of reads from biological replicates, under the cutoff of Benjamini–Hochberg adjusted *p*-value less than 0.05 and absolute fold-change larger than 1.5. The impulse-based differentially expressed genes (iDEGs) were called using the case-control mode of the Bioconductor package ImpulseDE2[22]. We added a pseudo-0 h time point using the 0.25 h UW data for the iDEGs analysis.

The highly expressed iDEGs (hiDEGs) were defined using the following criteria: (1) in the top 75% of iDEGs sorted by the sum of median gene expression values under W and UW conditions in decreasing order; (2) the W/UW fold change of median expression values being larger than two in at least one time point.

Wound-induced genes were defined from the hiDEGs by using a more strict criterion of requiring statistically significant two-fold up-regulation in at least one time point based on DESeq2 with an FDR cutoff at 0.05 in comparing W to UW samples.

### Transcription factor (TF)-gene regulatory network construction
The binding peaks of 283 transcription factors (TFs) were downloaded from the ENCODE portal, including 590 optimal IDR thresholded narrow peak files[48]. The 400 binding motifs of 371 *C. elegans* TFs were downloaded from the CIS-BP database[49]. The fimo program[50] was used to search the TF motif in the gene promoter region (within 2 kb upstream or 1 kb downstream of the transcription start site, TSS) under default parameters. The target genes of a TF were identified by requiring the presence of its ChIP-seq peak(s) or motif(s) within the gene promoter regions.

## Construction of TF dynamic regulatory network

TF dynamic regulatory network was constructed using Interactive visualization of dynamic regulatory networks (iDREM) program based on an HMM model with default parameters except for the Minimum_Standard_Deviation parameter set to 1.0, using the time-course expression data and the TF-target gene matrix described above[25]. The time-course expression data is a $\log_2$-transformed gene by time point matrix of the fold-change values of the median counts in wounded samples compared to those in unwounded samples for 3366 hiDEGs and 13 time points (including 0 h). We filtered out the TFs that were not expressed at the regulating time point with a cutoff of FPKM > 1 from the network. A split path in the dynamic regulatory network generated by iDREM represents a divergence of genes that are co-regulated up to that time point. We used all the paths before 10 h.p.w. for the time-course expression clustering analysis to identify different patterns of temporal gene expression during wound repair.

Not all expressed C. elegans TFs were included in the iDREM analysis, and some expressed genes without expression changes were included. Briefly, there are 587 expressed TFs and 310 expressed cofactors in total. We used all the TFs/cofactors with public ChIP-seq data or motif information (not requiring expression change), including 518 TFs and 5 cofactors, as input for iDREM analysis, followed by filtering non-expressed genes at the regulated time points. The final gene regulatory network we constructed has 237 TFs and 4 cofactors.

## Differential expression analysis of mouse bulk RNA-seq data

The correspondence between mouse and nematode homologous genes was derived from the wormbase (https://wormbase.org). The reads counts of all genes were downloaded from the GEO database under accession number GSE178758, and were then used for statistical analysis by DESeq2[47].

## TF-chromatin factor interaction analysis

All C. elegans chromatin factors were obtained from the wormbook database[51], and the protein-protein interactions between TF and chromatin factor were retrieved from the wormbase (https://wormbase.org). The interactions were visualized using the Cytoscape program[52].

## Identification of TF regulatory modules

We used the R package topicmodels to identify combinatory TF modules (Topics) based on the regulatory networks of TF-genes identified from iDREM. Here, in topic model terminology, each targeted gene was considered as a "document" while its regulating TFs were considered as "words", and a combination of TFs that co-regulate multiple genes were defined as a module or topic. The regulatory network at time point $I$ is converted into a gene-TF matrix $M_i$, and all the matrices at 0.25, 0.5, 6, 14, and 24 h.p.w. were horizontally concatenated together by adding a time point tag to the TFs (the same TF at different time points were treated as different TFs). The concatenated matrix $M$ was used as input, and an optimal number of 8 topics was estimated by using the four available metrics with the Gibbs method in the topicmodels package. A TF was considered to participate in a topic when the z-score of the TF-topic pair is larger than 2, and similarly, a target gene was considered to participate in a topic when the z-score of the gene-topic pair is larger than 1, following the same criteria as before[31].

The protein-protein interactions between the TFs in each topic were analyzed using the R package STRINGdb[53]. The microarray expression data of wildtype and daf-16 or jun-1 RNAi conditions were downloaded from the GEO database under accession GSE27677.

## TF intrinsically disordered regions (IDRs) analysis

The intrinsically disordered regions of TFs were downloaded from MobiDB, in which the IDR predictions were generated by MobiDB-lite software[54].

## GO enrichment analysis

Enriched GO-terms in DEGs were called using the Bioconductor package clusterProfiler with default parameters and a cutoff at 0.05 for adjusted $p$-values[55].

## Statistics and reproducibility

The statistical methods applied in all analyses are described in the corresponding sections. All the statistical tests were performed using R packages tailored to each analysis.

## Reporting summary

Further information on research design is available in the Nature Portfolio Reporting Summary linked to this article.

## Data availability

The raw sequencing data from this study have been deposited in the Genome Sequence Archive[56] in National Genomics Data Center[57], China National Center for Bioinformation / Beijing Institute of Genomics, Chinese Academy of Sciences (GSA: CRA008476 and CRA014914) that are publicly accessible at https://ngdc.cncb.ac.cn/gsa. All source data are deposited in FigShare with https://doi.org/10.6084/m9.figshare.25608453.

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

## Acknowledgements

This work was supported by grants from the National Natural Science Foundation of China (82341023 and 31922039 to Y.Z., 31972891 to S.X., and 32200682 to J.X.), the Ministry of Science and Technology of China

(2023YFC2307802) and the Fundamental Research Funds for the Central Universities (2042022dx0003) to Y.Z., the National Key R&D Program of China (2021YFA1300302, 2021YFA1101002). Part of the computation in this work was done on the supercomputing system in the Supercomputing Center of Wuhan University.

## Author contributions

Y.Z., S.X., and X.Y. conceived the study. J.Z., W.Y., J.X., R.L., and L.H. performed the experiments. X.Y. performed the analysis of the sequencing data, and M.W. validated the computational pipeline. X.Y., Y.Z., and S.X. wrote the manuscript with input from W.Y. and Y.C. All authors discussed the results and approved the manuscript.

## Competing interests

The authors declare no competing interests.
