## [Peer review file · Communications Biology]

Reviewers' comments:

Reviewer #3 (Remarks to the Author):

In this revision the authors have added extensive data validating the core swRNA-seq data that further strengthens confidence in the data quality. In several places the authors go in great detail about how various pieces of evidence indicate "great data quality" or similar things – I suggest removing these as they are distracting and it is enough to say "experiment Y validates the results of experiment X." I have a number of remaining minor issues that should be addressed before publication (listed below).

elt-2 is identified as a "negative regulator" of wound healing – given the well characterized role of this gene in the intestine, not epidermis, this is worth commenting on.

Many minor grammatical issues remain – I highlight some from the first few pages here but similar issues remain throughout.

4:15 should be "nematodes HAVE a remarkable repair capacity"

4:19 missing "and" before "recruitment"

5:12 grammar issue and also to address the past swRNA-seq studies noted by other reviewers, it would be better to say "we developed an IMPROVED single-worm RNA sequencing method (swRNA-seq) and USED THIS METHOD TO EXAMINE ..."

5:17 "revealed three interconnected stages WE TERMED response"

Currently nomenclature is NOT correct – many gene names are not italicized and use of capitals to indicate protein is not consistent. Please refer to this reference for details:

http://www.wormbook.org/chapters/www_nomenclature/caenornomenclature.html

6:9 – I think this is too dismissive of the past swPCR efforts which were impressive in their own right. Better to say something like "we developed a swPCR approach that improves on past methods (cite) by XYZ"

Paragraph starting 7:24 – distracting to have this much explanation; enough I think to say that differences in synchrony between bulk and sw could explain some of the differences...

Paragraph starting 8:3 – again I think the key point here really should be that while there may be power differences, the direction of effect is highly consistent between bulk and swRNA-seq which provides confidence that the DE genes identified are robust

8:15 remove word "meticulously", and remove unnecessary superlative "great quality" from end of this paragraph

8:25 missing space after "h.p.w"

12:23: "the top 15 TFs...including...wnt7b" – needs to still be reworded more since wnts are not TFs

16:26- clarify in the text whether these mutant RNA-seq experiments were done after wounding

Reviewer #4 (Remarks to the Author):

This ms analyzes the gene expression changes following skin/epidermal wounding in the nematode *C. elegans*. *C. elegans* is an excellent model for wound healing/repair responses and there have been several previous studies of transcriptional changes after wounding, e.g. in innate immune response. This work extends and adds to prior studies by performing a high temporal resolution (12 time points over 24 h) time course analysis of the wound response by single-worm RNAseq

methods, which may improve over bulk RNAseq. The authors report that ~9000 genes are differentially regulated at one or more time points, involving a complex regulatory network of >200 transcription factors acting in multiple phases. They perform functional validation experiments on selected transcription factors, which support a role in wound repair.

Overall the ms is a valuable addition to the wound healing and transcriptomics literature. The authors have done a lot of analysis and the data sets should be of use to other workers in the field. Whether the wound response involves such a large number of factors may be questioned. The authors suggest about half of all *C. elegans* TFs and almost half of all the genes are regulated, but this may reflect that the default thresholds that seem to be used are not stringent enough for this experimental design.

The only issue I would prefer the authors address is the accessibility of the data. While raw data are in an online repository and table S2 gives a list of DEGs (defined as altered at ANY timepoint), it would be more helpful to show a supp table of all genes broken down by expression at each time points. If this was somewhere else I was not able to find it. This would help readers trying to reproduce some of the analyses e.g. the circos diagram in Fig 1. While the raw data are present on the BIG data center repository, I recommend the source data for all specific graphs and plots shown in the ms should be explicitly provided.

Minor points:

In Page 12, Line 28; Page 13, Line 1 - Number of TFs and target genes are not the same as in Figure 3D which is also misspelled as Fig6D (Page 13, Line 2). E.g. text states "227, 143, and 142" TFs but Figure 3D shows 206, 120, and 121. Moreover, it is not clear from supplementary data set in Table S3 gene lists which TFs are in which category. The specific lists should be presented as part of supplemental data.

There are lot of gene names that aren't italicized throughout the manuscript. It would be good for the authors to follow standard *C. elegans* nomenclature in italicizing gene names. For example: Page 12, Line 23 *cwn-1*; Page-15, Line 8,9 *nhr-28*, *pqm-1*, *nhr-178*, *fos-1*, and *jun-1* Line-15 *daf-16*, *nhr-28*, *snpc-4* and *skn-1* line27 *jun-1* and *pqm-1*, etc.

Reviewer #5 (Remarks to the Author):

The authors have produced a valuable work on the transcriptional response to wound healing, which has been further enhanced by their response to previous reviewers. However several small issues still remain to be resolved in the manuscript text and figure legends.

1. The response to Reviewer #3's concern that inferred regulatory relationships from in silico analysis are presented as bona fide targets does not go far enough. Inferred targets are still presented as targets in pg.14 lines 4, 8-11, and figure legend for Fig 4, just to name a few examples. Every time the word target is used it should be paired with moderating language, like "putative" or "inferred", unless loss of function experiments or other validation have been performed as in Figure 6.

2. The results from the qRT-PCR analysis have not been appropriately integrated into the text. The qRT-PCR data shows that *nhr-76* and *daf-16* mRNA levels are not changed in response to wounding. However, the other data the authors present show these TFs change in nuclear localization and play clear roles in wound healing. I don't think it is particularly surprising that the response is post-transcriptional.

3. It is unclear whether all expressed *C. elegans* TFs were included in the iDREM analysis or if only those that changed in expression level were analyzed.

Minor points:

4. *Wnt7b* and *cwn-1* are Wnt ligands, not transcription factors.

5. Fig 4 only shows ChIP-seq peaks for the putative autoregulatory TFs. Do these also co-occur

with motifs? This would provide additional evidence for autoregulation.

Reviewer comments (Our point-by-point responses are in blue):

Reviewer #3:

In this revision the authors have added extensive data validating the core swRNA-seq data that further strengthens confidence in the data quality.

Response: Thanks for the positive comments on our study.

In several places the authors go in great detail about how various pieces of evidence indicate “great data quality” or similar things – I suggest removing these as they are distracting and it is enough to say “experiment Y validates the results of experiment X.”

Response: We have revised the text as suggested below to avoid distraction.

- “Together, RT-qPCR validates the gene expression changes observed in swRNA-seq results.”, on Page 8, Line 17.
- “Together, these results confirm previous reports, providing a basis for authentic identification of functional genes involved in wound repair.”, on Page 8, Line 27.

I have a number of remaining minor issues that should be addressed before publication (listed below).

elt-2 is identified as a “negative regulator” of wound healing – given the well characterized role of this gene in the intestine, not epidermis, this is worth commenting on.

Response: Thanks for the suggestion. We have added the comments below in the Discussion section of the revised manuscript.

“A recent study indicates that ELT-2 can function as a negative regulator in the intestine during embryo development in *C. elegans*, in which ELT-2 has been found to repress its expression across multiple developmental stages, as demonstrated by reporter experiments³⁶. Interestingly, a previous study shows that ELT-2 activates its mRNA expression when exposed to heat shock, as evidenced by promoter-reporter experiments³⁷. Another report highlights ELT-2’s essential role in early immune responses against pathogens and recovery from *S. enterica* infection³⁸. Here, our findings suggest that ELT-2 plays similar roles in the epidermis during wound repair, potentially forming an autoregulatory loop and repressing the expression of its target genes. We observed

inhibition of epidermal wound repair following *elt-2* RNAi knockdown. These results suggest that ELT-2 has broad functions during development and under stress conditions, probably through interactions with other transcriptional factors, including chromatin factors such as MRG-1 (Fig. S8D).”, on Page 21, Line 3.

Many minor grammatical issues remain – I highlight some from the first few pages here but similar issues remain throughout.

Response: Thanks for the suggestions. We have revised the whole manuscript accordingly.

4:15 should be “nematodes HAVE a remarkable repair capacity”

Response: Corrected.

4:19 missing “and” before “recruitment”

Response: Added.

5:12 grammar issue and also to address the past swRNA-seq studies noted by other reviewers, it would be better to say “we developed an IMPROVED single-worm RNA sequencing method (swRNA-seq) and USED THIS METHOD TO EXAMINE ...”

Response: Thanks, and we have revised the sentence as suggested: “We developed an **improved** single-worm RNA sequencing (swRNA-seq) method, and **used it to examine ...**” on Page 5, Line 12.

5:17 “revealed three interconnected stages WE TERMED response

Response: Added.

Currently nomenclature is NOT correct – many gene names are not italicized and use of capitals to indicate protein is not consistent. Please refer to this reference for details: http://www.wormbook.org/chapters/www_nomenclature/caenornomenclature.html

Response: Thanks, and we have corrected the nomenclature errors by revising all gene names to be italicized and protein names to be capitalized in the revised manuscript.

6:9 – I think this is too dismissive of the past swPCR efforts which were impressive in their own right. Better to say something like “we developed a swPCR approach that improves on past methods (cite) by XYZ”

Response: Thanks, and we have revised the original sentence as suggested.

Paragraph starting 7:24 – distracting to have this much explanation; enough I think to say that differences in synchrony between bulk and sw could explain some of the differences...

Response: As suggested, we have simplified the original three sentences to “Moreover, we reason that swRNA-seq has an advantage in synchrony over bulk RNA-seq, especially for the earlier time-point after wounding.” on Page 7, Line 27.

Paragraph starting 8:3 – again I think the key point here really should be that while there may be power differences, the direction of effect is highly consistent between bulk and swRNA-seq which provides confidence that the DE genes identified are robust

Response: As suggested, we have added the point that “While there are some power differences, the direction of effect is highly consistent between swRNA-seq and bulk RNA-seq, which provides confidence that DEGs identified are robust” at the end of this paragraph on Page 8, Line 10.

8:15 remove word “meticulously”, and remove unnecessary superlative “great quality” from end of this paragraph

Response: Corrected.

8:25 missing space after “h.p.w”

Response: Fixed.

12:23: “the top 15 TFs...including...wnt7b” – needs to still be reworded more since wnts are not TFs.

Response: Thanks for pointing it out. As suggested, we have revised the original sentence: “the top 15 TFs/**cofactors** ..., including *jun-1*, *fos-1*, *cwn-1*, etc”. We examined both transcription factors (TFs) and transcription cofactors including *cwn-1*, a homologous gene of Wnt7b in mice. We have revised Figure 3C and legend accordingly on Page 12, Line 20.

16:26- clarify in the text whether these mutant RNA-seq experiments were done after wounding.

Response: Yes, these mutant RNA-seq experiments were done after wounding. We have added this information in the revised sentence: “We selected two TFs, *jun-1* and *pqm-1*, as representatives to perform RNA-seq experiments **after wounding** upon the loss of a single TF and both TFs” on Page 16, Line 27.

Reviewer #4:

This ms analyzes the gene expression changes following skin/epidermal wounding in the nematode *C. elegans*. *C. elegans* is an excellent model for wound healing/repair responses and there have been several previous studies of transcriptional changes after wounding, e.g. in innate immune response. **This work extends and adds to prior studies** by performing a high temporal resolution (12 time points over 24 h) time course analysis of the wound response by single-worm RNAseq methods, which may improve over bulk RNAseq. The authors report that ~9000 genes are differentially regulated at one or more time points, involving a complex regulatory network of >200 transcription factors acting in multiple phases. They perform functional validation experiments on selected transcription factors, which support a role in wound repair.

Response: Thanks for the positive comments and summary of our study.

Overall the ms is a valuable addition to the wound healing and transcriptomics literature.

The authors have done a lot of analysis and the data sets should be of use to other

workers in the field. Whether the wound response involves such a large number of factors may be questioned. The authors suggest about half of all *C. elegans* TFs and almost half of all the genes are regulated, but this may reflect that the default thresholds that seem to be used are not stringent enough for this experimental design.

Response: We acknowledge the reviewer's concern regarding the complexity of factors involved in the wound response. "Wound repair is one of the most complex processes that occur during human life" (Gurtner et al. Nature 2008). Here, we tracked 12 time points spanning 24 hours during wound repair in *C. elegans*, which entails a dynamic interplay of many factors involved in different molecular events, such as actin reorganization, membrane trafficking, protein recruitment, membrane fusion, and remodeling, all unfolding in a time-dependent manner. We identified 241 TFs/cofactors (22.6%) out of 1065 factors in *C. elegans* (Animal TF database 3.0). These putative factors and their targets merit further validation and could be used as candidates for researchers in relevant fields. We have added a note of this point to the Discussion on Page 20, Line 3.

"Of note, we used the default thresholds in aiming to identify all putative factors and to establish a complete map for further studies. More stringent cutoffs can be used to identify the factors with more significant changes, and experimental validations are necessary to identify direct regulators and understand their functional roles in wound repair."

The only issue I would prefer the authors address is the accessibility of the data. While raw data are in an online repository and table S2 gives a list of DEGs (defined as altered at ANY timepoint), it would be more helpful to show a supp table of all genes broken down by expression at each time points. If this was somewhere else I was not able to find it. This would help readers trying to reproduce some of the analyses e.g. the circos diagram in Fig 1. While the raw data are present on the BIG data center repository,. I recommend the source data for all specific graphs and plots shown in the ms should be explicitly provided.

Response: Thanks for this suggestion. We have provided a large expression table of all genes by time points and wound/unwound conditions (exp.tsv) deposited in FigShare (DOI: 10.6084/m9.figshare.25608453). The source data for all specific graphs and plots in the revised manuscript are also deposited in the same FigShare repository.

Minor points:

In Page 12, Line 28; Page 13, Line 1 - Number of TFs and target genes are not the same as in Figure 3D which is also misspelled as Fig6D (Page 13, Line 2). E.g. text states “227, 143, and 142” TFs but Figure 3D shows 206, 120, and 121. Moreover, it is not clear from supplementary data set in Table S3 gene lists which TFs are in which category. The specific lists should be presented as part of supplemental data.

Response: Thanks for the suggestions. We have revised the numbers of TFs to “206, 120, and 121” in Fig. 3D, and corrected the misspelled Fig. 6D to Fig. 3D. For Table S3 gene list, the gene category by stage at which it works (Response, Repair, and Remodeling) is annotated in the column “Stage”. We have also provided the three stage-specific gene lists in the source data of Fig. 3D, deposited in the same FigShare repository (DOI: 10.6084/m9.figshare.25608453) as above.

There are lot of gene names that aren't italicized throughout the manuscript. It would be good for the authors to follow standard *C. elegans* nomenclature in italicizing gene names. For example: Page 12, Line 23 *cwn-1*; Page-15, Line 8,9 *nhr-28*, *pqm-1*, *nhr-178*, *fos-1*, and *jun-1* Line-15 *daf-16*, *nhr-28*, *snp-4* and *skn-1* line27 *jun-1* and *pqm-1*, etc.

Response: We have standardized all the nomenclature throughout the manuscript as suggested, with gene names now appropriately italicized.

Reviewer #5:

The authors have produced a valuable work on the transcriptional response to wound healing, which has been further enhanced by their response to previous reviewers. However several small issues still remain to be resolved in the manuscript text and figure legends.

Response: Thanks for the positive comments on our study.

1. The response to Reviewer #3's concern that inferred regulatory relationships from in silico analysis are presented as bona fide targets does not go far enough. Inferred targets are still presented as targets in pg.14 lines 4, 8-11, and figure legend for Fig 4, just to name a few examples. Every time the word target is used it should be paired with

moderating language, like "putative" or "inferred", unless loss of function experiments or other validation have been performed as in Figure 6.

Response: Thanks for the suggestion. We have added “putative” or “inferred” before the “target” when validation is not performed throughout the revised manuscript.

2. The results from the qRT-PCR analysis have not been appropriately integrated into the text. The qRT-PCR data shows that *nhr-76* and *daf-16* mRNA levels are not changed in response to wounding. However, the other data the authors present show these TFs change in nuclear localization and play clear roles in wound healing. I don't think it is particularly surprising that the response is post-transcriptional.

Response: As suggested, we have integrated the RT-qPCR results into the revised text described below.

“This *daf-16* result was further validated by RT-qPCR, consistent with the result of swRNA-seq (Fig. S3), suggesting a post-transcriptional regulation.” (Page 18, Line 14)

“Similarly, *nhr-76* has little expression change at 0.5 h (Fig. S3) but protein translocation from cytoplasm to the nucleus.” (Page 18, Line 25)

3. It is unclear whether all expressed *C. elegans* TFs were included in the iDREM analysis or if only those that changed in expression level were analyzed.

Response: Not all expressed *C. elegans* TFs were included in the iDREM analysis, and some expressed genes without expression changes were included. Briefly, there are 587 expressed TFs and 310 expressed cofactors in total. We used all the TFs/cofactors with public ChIP-seq data or motif information (not requiring expression change), including 518 TFs and 5 cofactors, as input for iDREM analysis, followed by filtering non-expressed genes at the regulated time points. The final gene regulatory network we constructed has 237 TFs and 4 cofactors. The procedure and the numbers of TFs and cofactors have been added to the Methods section in the revised manuscript (Page 28, Line 23).

Minor points:

4. *Wnt7b* and *cwn-1* are Wnt ligands, not transcription factors.

Response: Thanks for the comment. In the revised manuscript, we have revised the description of *cwn-1* in *C. elegans* and *Wnt7b* in mice as transcription cofactors.

5. Fig 4 only shows ChIP-seq peaks for the putative autoregulatory TFs. Do these also co-occur with motifs? This would provide additional evidence for autoregulation.

Response: Thanks for the suggestion. For target gene *nhr-178* in Fig. 4a, unfortunately, no ChIP-seq data is available for TF NHR-178. For target gene *jun-1* in Fig. 4b, the ChIP-seq data for JUN-1 are from L1, L3, and L4 larva *C. elegans* and we did not find ChIP-seq peaks co-occurring with motifs. This is probably because the ChIP-seq data is not from the right developmental stage or wound stress condition. Further studies by performing NHR-178 and JUN-1 ChIP-seq in parallel with swRNA-seq upon wound repair are necessary to investigate their autoregulation. We have marked the autoregulatory TFs as “putative”, and added a discussion of requiring ChIP-seq in future studies in the revised manuscript (Page 21, Line 2).

REVIEWERS' COMMENTS:

Reviewer #3 (Remarks to the Author):

For the most part I'm satisfied with the authors' changes.

However I remain highly skeptical about the framing of the elt-2 results. The new text suggesting "elt-2 plays a similar role in the epidermis" is not supported by extensive prior studies that elt-2 is NOT in fact expressed in the epidermis. Much more likely this reflects a non-cell-autonomous effect, where elt-2 in the intestine alters some thing (metabolism etc) that then impacts wound healing in the epidermis. Without clear data showing elt-2 is in fact acting in the epidermis this text needs to be revised.

Reviewer #4 (Remarks to the Author):

The revised manuscript has fully addressed my concerns and I support publication.

Reviewer #5 (Remarks to the Author):

The changes made by the authors have greatly clarified and improved this manuscript. However, one minor error remains to be corrected. The change made in response to the comments from reviewers 3 & 5 that Wnt ligands cwn-1/Wnt7b are not transcription factors is still technically incorrect. These proteins are not considered transcriptional co-factors because they never enter the nucleus and do not directly participate in transcription. It would be more appropriate to refer to them as "signaling proteins" or "extracellular signaling molecules" or something to that effect.

Reviewer comments (Our point-by-point responses are in blue):

Reviewer #3:

For the most part I'm satisfied with the authors' changes.

However I remain highly skeptical about the framing of the elt-2 results. The new text suggesting "elt-2 plays a similar role in the epidermis" is not supported by extensive prior studies that elt-2 is NOT in fact expressed in the epidermis. Much more likely this reflects a non-cell-autonomous effect, where elt-2 in the intestine alters some thing (metabolism etc) that then impacts wound healing in the epidermis. Without clear data showing elt-2 is in fact acting in the epidermis this text needs to be revised.

Response: Thanks. As suggested, we have removed “in the epidermis” on Page 21, Line 21, and added “probably mediated through a non-cell autonomous effect” in the Discussion on Page 21, Line 25.

Reviewer #4:

The revised manuscript has fully addressed my concerns and I support publication.

Reviewer #5:

The changes made by the authors have greatly clarified and improved this manuscript.

However, one minor error remains to be corrected. The change made in response to the comments from reviewers 3 & 5 that Wnt ligands cwn-1/Wnt7b are not transcription factors is still technically incorrect. These proteins are not considered transcriptional co-factors because they never enter the nucleus and do not directly participate in transcription. It would be more appropriate to refer to them as "signaling proteins" or "extracellular signaling molecules" or something to that effect.

Response: Thanks. As suggested, we have revised “TFs/cofactors” to a more general term, “transcription regulators (TFs, cofactors, or signaling proteins),” and referred to Wnt7b as “signaling protein” in the revised manuscript, on Page 12, Line 28 - Page 13, Line 2.